# Application of network analysis and cluster analysis for better prevention and control of swine diseases in Argentina

**Jerome N. Baron** [1]*, **Maria N. Aznar**[2], **Mariela Monterubbianesi**[3], **Beatriz Martínez-López**[1]

**1** Department of Medicine and Epidemiology, School of Veterinary Medicine, Center for Animal Disease Modeling and Surveillance (CADMS), University of California Davis, Davis, California, United States of America, **2** Instituto Nacional de Tecnología Agropecuaria (INTA), Buenos Aires, Argentina, **3** Servicio Nacional de Sanidad y Calidad Agroalimentaria de la Republica Argentina (SENASA), Buenos Aires, Argentina

* jnbaron@ucdavis.edu

## Abstract

### Rationale/background

Though much smaller than the bovine industry, the porcine sector in Argentina involves a large number of farms and represents a significant economic sector. In recent years Argentina has implemented a national registry of swine movements amongst other measures, in an effort to control and eventually eradicate endemic Aujesky's disease. Such information can prove valuable in assessing the risk of transmission between farms for endemic diseases but also for other diseases at risk of emergence.

### Methods

Shipment data from 2011 to 2016 were analyzed in an effort to define strategic locations and times at which control and surveillance efforts should be focused to provide cost-effective interventions. Social network analysis (SNA) was used to characterize the network as a whole and at the individual farm and market level to help identify important nodes. Spatio-temporal trends of pig movements were also analyzed. Finally, in an attempt to classify farms and markets in different groups based on their SNA metrics, we used factor analysis for mixed data (FAMD) and hierarchical clustering.

### Results

The network involved approximate 136,000 shipments for a total of 6 million pigs. Over 350 markets and 17,800 production units participated in shipments with another 83,500 not participating. Temporal data of shipments and network metrics showed peaks in shipments in September and October. Most shipments where within provinces, with Buenos Aires, Cordoba and Santa Fe concentrating 61% of shipments. Network analysis showed that markets are involved in relatively few shipments but hold strategic positions with much higher betweenness compared to farms. Hierarchical clustering yielded four groups based on SNA metrics and node characteristics which can be broadly described as: 1. small and backyard

**Data Availability Statement:** Data cannot be shared publicly as this data is owned by a third-party (the National Service of Agri-Food Health and

Quality of the Argentine Government, SENASA) and has confidentiality issue as this is individual census data. Data accessibility and restriction information can be obtained from the National Service of Agri-Food Health and Quality (SENASA). For information about data accessibility and data requests please contact infopublica@senasa.gob.ar. The authors confirm that they had no special privileges to the data and that other researchers will be able to access the data in the same manner as the authors.

**Funding:** The authors received no specific funding for this work.

**Competing interests:** The authors have declared that no competing interests exist.

farms; 2. industrial farms; 3. markets; and 4. a single outlying market with extreme centrality values.

## Conclusion

Characterizing the network structure and spatio-temporal characteristics of Argentine swine shipments provides valuable information that can guide targeted and more cost-effective surveillance and control programs. We located key nodes where efforts should be prioritized. Pig network characteristics and patterns can be used to create dynamic disease transmission models, which can both be used in assessing the impact of emerging diseases and guiding efforts to eradicate endemic ones.

## 1. Introduction

The porcine sector in Argentina is a relatively small industry comparatively to the beef sector, representing only about 2% of the Argentinian livestock population [1]. This is similar to its neighboring countries of Paraguay and Uruguay also covering the great plains of the South-Eastern America. However, Argentinian swine production includes a robust industrial sector as well as numerous backyard farmers whose livelihood depend on the small number of animals they raise. Argentina has established goals to eradicate endemic diseases, with high economic costs, such as Aujesky's disease which has been present in the country since 1979 [2, 3] and to prevent the introduction of others for which the country is free, such as porcine reproductive respiratory syndrome (PRRS), African swine fever (ASF) and classical swine fever (CSF) [3]. In Argentina all movements of domestic livestock must be declared to the state veterinary service (National Service for Agrifood Health and Quality, SENASA). If statutory requirements are met, SENASA allows the movement of identified animals by issuing a permit and data are recorded and stored in a database called the Integrated System of Management in Animal Health (Sistema Integrado de Gestión de Sanidad Animal, Sigsa). Many infectious agents are mainly transmitted between farms through the transport of live animals or via contaminated fomites carried by vehicles such as trucks transporting animals or products [4]. Thus, the analysis of pig movement networks can provide valuable insights to design more cost-effective risk-based surveillance and control programs for diseases for which the country aims to achieve eradication, like Aujesky's disease. Moreover, with the global re-emergence of diseases such as PRRS or ASF, it may help to better prevent and potentially control any of those transboundary, diseases if they enter the country.

The use of social network analysis (SNA) and graph theory has been used in multiple instances to characterize animal movements within a given livestock sector. This has been used extensively to characterize movement networks for swine in Europe [5–7] and more recently in North America and other regions [8, 9]. In South America, the method has been used to characterize cattle movements in Uruguay [10] and Argentina [11] but to the best of our knowledge has been scarcely used in the swine industry to date. In combination with other methods such as mapping [12], epidemic simulation using the network structure [13] and space-time clustering [14], SNA can define locations in time and space that are strategic for the implementation of surveillance programs by for example, identify major nodes that can act as super-spreaders and super-receivers, or identify communities and other network structures that may be used to prevent disease transmission among regions or maximize the effectiveness of control and vaccination programs.

The objective of this study is to describe and characterize the spatio-temporal swine movement network in Argentina. For such purpose we will use a combination of spatio-temporal analysis methods, network analysis and unsupervised machine learning techniques (cluster analysis). Results of this study would inform the design of more cost-effective prevention and control programs for swine diseases in the country and contribute to swine production improvement and sustainability in South America.

## 2. Methods

### 2.1. Data collection and sources

In Argentina, the following data are recorded for each movement event: the province and district of origin, the unique identifier of the source farm or market (RENSPA) and its geolocation (latitude and longitude), the date animals are to be transported, the species involved, the number of individuals by age category, the reason for the movement, the province and district of destination and the RENSPA and geolocation of the destination premise (farm, market or slaughterhouse). These data are recorded and stored in a database called the Sanitary Management System (Sistema de Gestión Sanitaria, Sigsa).

The swine demographics and movement data were provided by SENASA. Two datasets were provided. The first one was the farm census of 2016, which included all registered productive units as defined by SENASA, with at least one pig on site, their geolocation, and the number of pigs (and other livestock species) in the unit. A productive unit is defined as the unit managed by one farmer; a single actual farm can contain multiple units if multiple farmers produce in the same farm. Therefore, the unit of observation in this study is the productive unit. The second dataset included all pig movements in Argentina between units, from/to markets, and to slaughterhouses from January 2011 to December 2016. For this study, shipments from units or markets to slaughterhouses were not included as they are considered dead-end points for disease transmission and we were particularly interested to focus our attention in the potential disease spread between farms.

### 2.2. Analysis

**2.2.1 Descriptive analyses and mapping.**   Spatio-temporal aspects of pig farming and movements in Argentina were described using tables, graphs, and maps. Bar plots were built on a monthly basis for overall movements, movements to and from markets, overall pigs moved, pigs moved through markets, and average shipment size to observe seasonal patterns.

Euclidean distance between shipping partners was computed using the geolocation of each unit or market from the dataset. Using these geolocations, units and pig movements were geographically mapped using the "maps" package in R [15, 16] which pulls its shapefiles from the open-source Natural Earth database [17].Points are plotted as units or markets involved in movements, and arcs as shipments. Maps of all pig shipments were created for the total 2011–2016 period as well as for each year and month. Similarly, maps with the subset of pig shipments involving markets were created. To improve visualization of areas with high density of swine farming and movements, kernel density maps were created in ArcGIS [18] for unit density, pig density, number of shipments per unit and number of pigs per shipment per unit.

**2.2.2 Network construction and visualization.**   The networks were built using the igraph package in R [19]. Nodes were defined as productive units and markets. Edges were defined as individual shipments and weighted using the number of pigs per shipment. We built directed networks, meaning the edges accounted for the direction of the shipment from one node to another (i.e., Unit A sends pigs to Unit B). Networks were created for the total dataset as well as by year to allow for comparison over time. Comparing networks over time help us to

understand if there are stable and predictable movement patterns and relationships. In this manner, we could identify specific nodes or groups of nodes that are likely to be important in future movements, and thus could be targeted as strategic points for surveillance and intervention strategies. Networks were graphed overall as well as on a monthly basis with a force-directed Kamada-Kawai layout [20] for better visualization of individual nodes as well as network structures. Graphing monthly networks allowed us to observe smaller structures where features could be better distinguished. Color-coding was used to define node type (productive unit or market).

**2.2.3 Network analyses.** From the full, yearly, and monthly networks we were able to determine how many units and markets were involved in pig movements as well as to compute key network metrics: in- and out-degree, betweenness, Eigen centrality, and network density. Closeness centrality could not be properly computed as this was a disconnected network [21, 22]. We examined weak and strong components to evaluate clustering. These measures, which have been previously described and shown relevant for preventive veterinary medicine [23], are briefly described in Table 1. With these metrics, it was possible to evaluate the global structure of the network, compare the roles of markets and units in the movement network, and evaluate the role of subgroups and individual nodes. Both weak and strong components allow the identification of groups of units that have an intensive trade relationship with each other. In terms of disease transmission, these components may help evaluate the extent to which an outbreak might spread, if started in a given location in the network [e.g. 10]. Individual unit and market metrics permit the evaluation of the level of activity and direct movements of an individual node (degree), as well as the position of the node in relation to the network (betweenness and Eigen centrality), which considers both direct and indirect connections. Individual nodes with outstanding values, thus holding strategic positions, could then be suitable for targeted intervention. For instance, in the case of an outbreak, it would be possible to determine which strategic nodes should be targeted first for surveillance and potential vaccination programs in a short period of time.

The sub-network without markets was also analyzed, given the apparent key role of markets in the network, to see how this would affect the cohesiveness of the network. The same metrics were measured for this sub-network. All analyses were conducted with R 3.3.1. [15] and mapped using ArcGIS 10.6.1 [18].

We also aimed to identify groups of units and markets with similar movement patterns. For such purpose we used FAMD (Factor Analysis for Mixed Data). FAMD is a variant of MFA

**Table 1. Definitions of social network centralities used in this study.**

| Metric | Definition | Reference |
|---|---|---|
| In/out-degree centrality | Total number of incoming or outgoing contacts during the period considered for a single node. This is a measure of the absolute connectivity of a given node | [23, 27, 28] |
| Betweenness centrality | For node A it is the sum of the proportion of shortest paths between pairs of other nodes in the network that go through node A. It's a measure of the importance node A has in connecting other nodes in the network which don't have a direct connection. | [23, 27, 29] |
| Eigen centrality | For a given node, it's centrality is a proportion of the sum of centralities of its neighbors | [27, 30] |
| Network density | Proportion of observed edges in the network compared to the total number of theoretical connections between all nodes. | [23, 31] |
| Strong component | Component considering direct connections only between nodes. Directionality of shipment is considered. | [22, 23, 32, 33] |
| Weak component | Component considering both direct and indirect connections between nodes. Directionality of shipment is considered. | [22, 23, 32, 33] |

(Multiple Factor Analysis) which can account for both categorical and continuous variables by combining PCA (Principal Component Analysis) for the continuous variables and MCA (Multiple Correspondence Analysis) for the categorical ones. For FAMD, two categorical variables (province and node type) and thirteen continuous variables were considered (unit area, unit population of pigs, unit population of other livestock, unit population of poultry, indegree, outdegree, betweenness, number of pigs shipped out, number of pigs received, average outgoing shipment size, average incoming shipment size, average distance of outgoing shipment, average distance of incoming shipment). Following the selection of a model, hierarchical clustering was used to define groups of nodes. Analysis was performed only on nodes which participated actively in pig movements at any given point during 2011 and 2016 and was conducted using the FactoMineR package in R [24].

**2.2.4 Missing data.** Some units and markets present in the shipment data (3.7% of the total nodes) were not present in the 2016 census, and lacked geolocation. In those cases, we used the mean values for longitude and latitude of other units in the same department as their locations. This approach was chosen over the department centroid as it assumed that units aren't always uniformly distributed within a department. In this case a unit with unknown coordinates is more likely to be closer to where other units might be concentrated or clustered.

## 3. Results

### 3.1 General characteristics of the pig industry in Argentina

The 2016 farm census recorded 97,605 productive units containing 4,988,169 pigs for 2016. These units also recorded 15,832,134 other large animals, including cattle, small ruminants and horses and 23,347,128 poultry. The average unit size was 51 pigs and the median was 8, with the largest unit registering 98,230 pigs.

In total, 739,786 movements were recorded between 2011 and 2016 involving 33,927,547 pigs. After taking out movements to slaughterhouses, analysis was performed on the remaining 135,538 movements for a total of 5,934,881 pigs involving farms and markets only. Average shipment size was 44 pigs, with the median being 20 pigs. The 75%, 95% and 99% percentiles were of 40, 141 and 450 pigs in a shipment. A total of 351 markets and 17,809 units were involved in recorded movements, forming 40,931 shipment pairs. The remaining 83,506 units from the census were not involved in shipping pigs between 2011 and 2016. The average number of shipments per pair of nodes was 3.3 and the median 1 (the 75%, 95% and 99% percentiles being of 2, 11 and 37 respectively, with the maximum shipments between a pair reaching 544). The average number of pigs shipped between pairs was 145, with a median of 15 (the 75%, 95% and 99% percentiles being of 56, 393 and 1498 pigs, with the maximum reaching 368,398 pigs shipped between a pair).

### 3.2 Temporal trends in pig movements

As shipment data collection was first introduced in 2011, seasonal trends could not be observed for that year, with the steady increase in the number of shipments during 2011 reflecting the increase in coverage and improvement in data collection, not an increase in shipments. Movement patterns showed that peak months in number of shipments for the period 2012 to 2016 were the months of September and October (average of 2,430 and 2,439 shipments, respectively) and the lowest months were January and February (average of 1,588 and 1,615) (Fig 1). Average monthly shipments for other months varied from 1,881 to 2,053. These observations were even more pronounced when looking at markets exclusively. When comparing to the average number of monthly shipments going through markets each year, September and October had 1.82 and 1.66 times the amount of shipments whereas January and

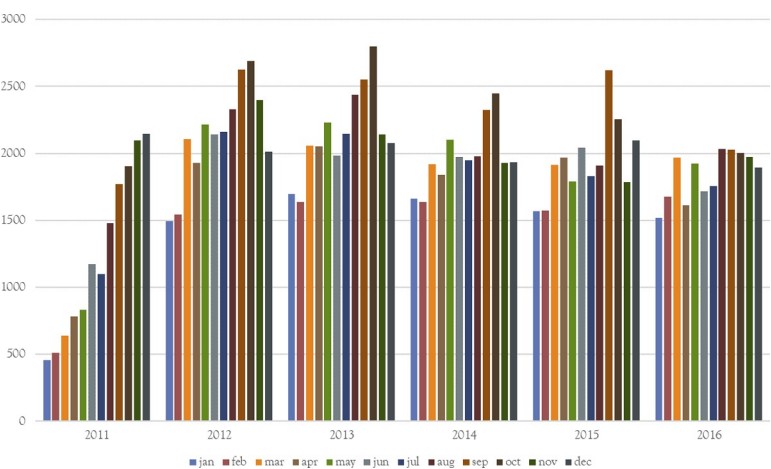

**Fig 1. Number of pig shipments per month between 2011 and 2016 in Argentina.**

February had 0.40 and 0.56 times the number of shipments. For other months these values varied between 0.79 and 1.16. However peak month for average shipment size were January and December (54 and 53 pigs per shipment), with the lowest months being September and October (39 and 40 pigs per shipment) with other months varying between 42 and 49 pigs per shipment on average. The average size of shipments increased steadily over the years from 34 pigs in 2011 to 55 in 2016 (median from 15 to 20) (Fig 2). Therefore, even though the number of shipments decreased from 25,655 in 2012 to 22,110 in 2016, the number of pigs shipped increased from 962,006 to 1,219,726. For the years 2012 to 2016, we observed very similar monthly patterns in movements, suggesting a relatively stable and predictable movement network in Argentina.

### 3.3 Spatial distribution of the swine industry in Argentina

Mapping swine movements from 2011 to 2016 shows a concentration of movements in the areas west of Buenos Aires, with a number of major actors on the periphery interacting with the core industrial center of swine production (Fig 3). In terms of unit and pig density, we can

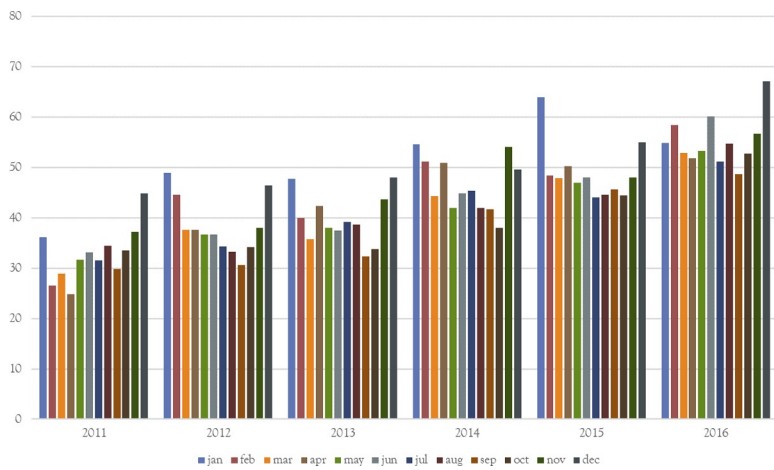

**Fig 2. Average shipment size per month between 2011 and 2016.**

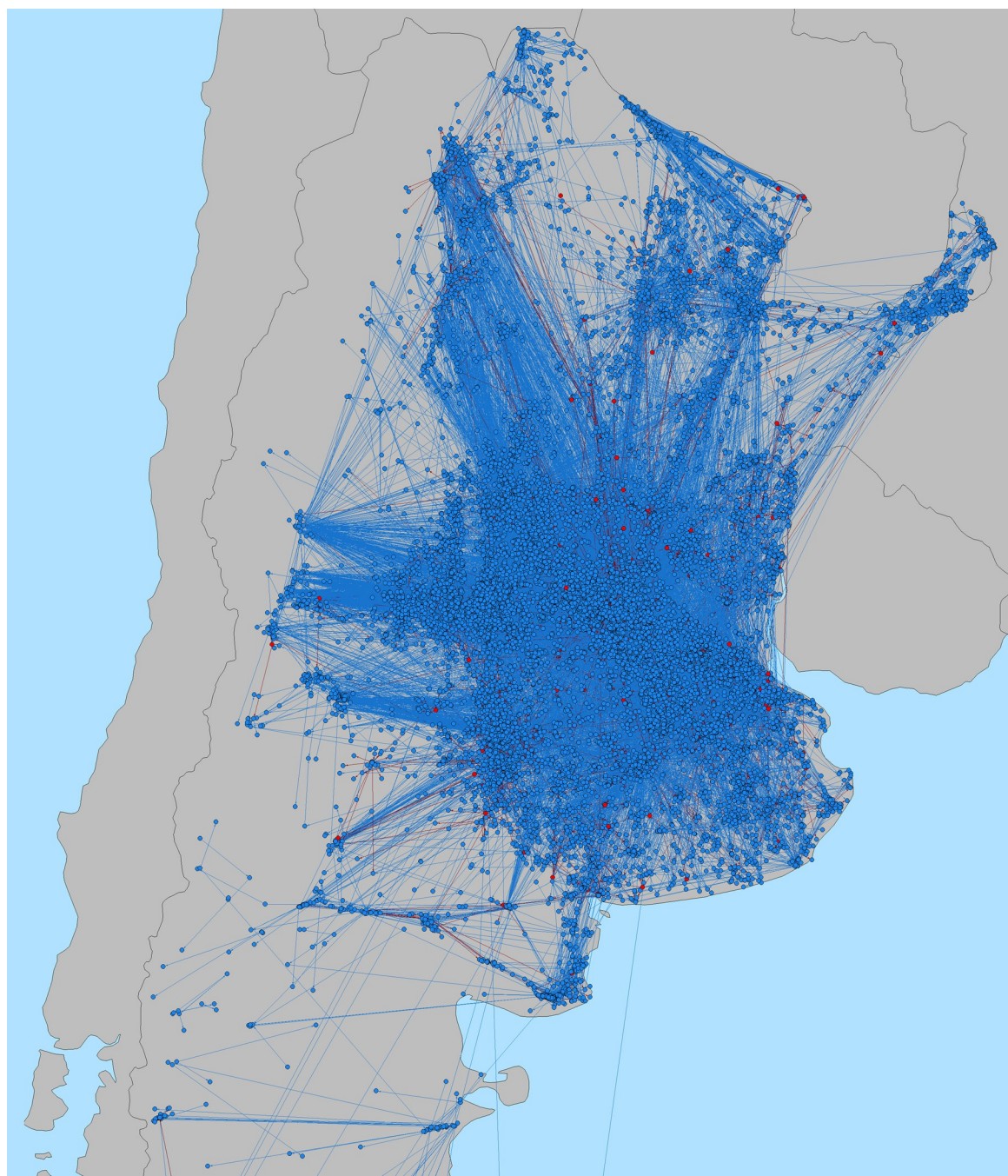

**Fig 3. Distribution of pig movements in Argentina from 2011 to 2016.** Red nodes represent markets and blues nodes farms. Red lines are movements coming from markets and blue lines coming from farms.

distinguish three distinct areas concentrated in the provinces of Formosa, Chaco, Corrientes and Misiones (Fig 4A–4B). The first is an area of high unit and high swine density covering parts of the provinces of Buenos Aires, Santa Fe and Cordoba, and to a lesser extent those of Entre Rios and San Luis. This area is on an axis that includes, from West to East, the cities of Buenos Aires, Rosario, Santa Fe, Cordoba and San Luis. Secondly, there is an area of high unit density but low swine density covering the Northeastern provinces of Formosa, Chaco and

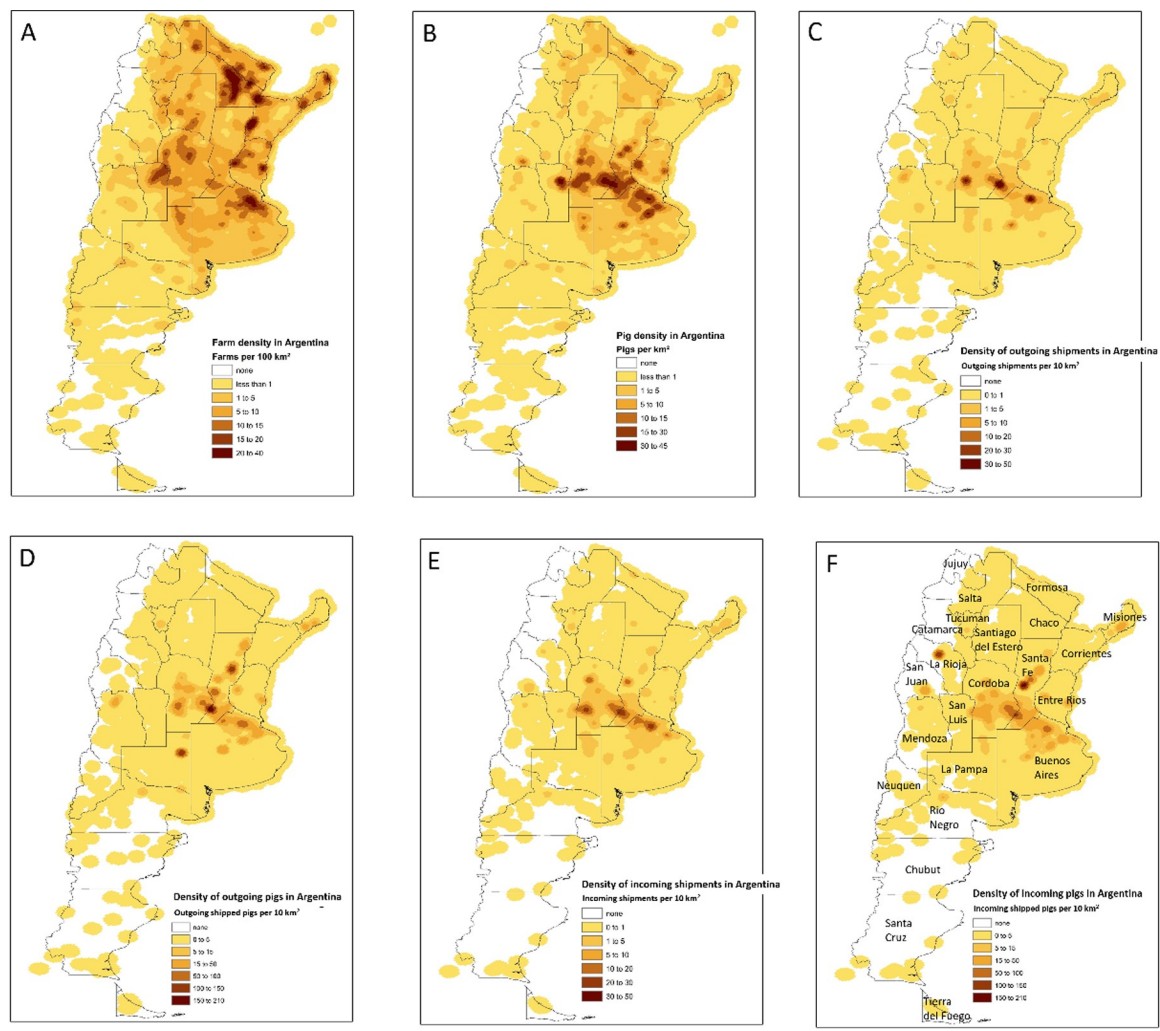

**Fig 4.** Kernel density maps: of Argentinian (A), farm distribution in 2016, (B) swine distribution in 2016, (C) outgoing shipments from 2011 to 2016, (D) outgoing traded pigs from 2011 to 2016, (E) incoming shipments from 2011 to 2016 and (F) incoming traded pigs from 2011 to 2016.

Misiones and to a lesser extent the provinces of Salta, Santiago del Estero and Corrientes. This area is essentially located along the border with Bolivia and Paraguay. Finally, the rest of the country has both low densities in units and pigs. At the provincial level, we can see that the 3 provinces of Buenos Aires, Cordoba and Santa Fe contain 36.6% of units (35,707) but 62.6% of pigs (3,123,567). The remaining 9 North-East provinces contain 57.0% of units (55,619) and 32.2% of pigs (1,606,938) and the 11 western and southern provinces contain 6.4% of units (6,279) and 5.2% of pigs (257,677) (Table 2). We observe that in all provinces, the mean size is always quite higher than the median, indicating strongly right-skewed distributions in productive unit sizes.

### 3.4 Spatial and provincial patterns of swine farming and movements

When looking at the density maps of swine movements and comparing them with swine population density, we confirm that there are similar spatial patterns with most movements being conducted around the industrial area of swine farming (Fig 4C–4F), with few major hot-spots

**Table 2. Farm and pig distribution by province in Argentina in 2016.**

| | Number of Productive units | Number of pigs | Average unit size | Median unit size | Largest unit |
|---|---|---|---|---|---|
| **Major swine producing provinces** | **35,707** | **3,123,567** | **87** | | |
| Buenos Aires | 17,762 | 1,226,498 | 69 | 10 | 56,910 |
| Cordoba | 12,017 | 1,117,913 | 93 | 12 | 24,281 |
| Santa Fe | 5,928 | 779,156 | 131 | 10 | 45,739 |
| **North East provinces** | **55,619** | **1,606,938** | **29** | | |
| Capital Federal | - | - | - | | |
| Chaco | 12,007 | 253,609 | 21 | 9 | 11,168 |
| Corrientes | 6,798 | 73,685 | 11 | 3 | 9,958 |
| Entre Rios | 6,224 | 345,370 | 55 | 6 | 13,883 |
| Formosa | 7,110 | 172,040 | 24 | 8 | 1,701 |
| La Pampa | 3,167 | 160,835 | 51 | 14 | 19,939 |
| Misiones | 3,669 | 65,591 | 18 | 5 | 2,998 |
| Salta | 6,185 | 220,586 | 36 | 17 | 3,673 |
| San Luis | 3,965 | 216,976 | 55 | 6 | 98,227 |
| Santiago del Estero | 6,494 | 98,246 | 15 | 7 | 2,694 |
| **Western and Southern provinces** | **6,279** | **257,677** | **41** | | |
| Catamarca | 1,080 | 15,732 | 15 | 4 | 943 |
| Chubut | 378 | 24,562 | 65 | 12 | 6,494 |
| Jujuy | 581 | 25,940 | 45 | 6 | 4,676 |
| La Rioja | 564 | 23,610 | 42 | 3 | 12,725 |
| Mendoza | 1,137 | 35,303 | 31 | 5 | 2,687 |
| Neuquen | 329 | 20,593 | 63 | 8 | 9,746 |
| Rio Negro | 805 | 31,205 | 39 | 9 | 5,210 |
| San Juan | 258 | 42,698 | 165 | 3 | 21,489 |
| Santa Cruz | 72 | 3,272 | 45 | 13 | 563 |
| Tierra del Fuego | 16 | 973 | 61 | 7 | 472 |
| Tucuman | 1,059 | 33,789 | 32 | 4 | 4,504 |
| **Total** | **97,605** | **4,988,182** | **51** | **8** | **98,227** |

concentrating most of the incoming and outgoing shipments. Of 135,538 shipments, 83,077 shipments (61.3%) were internal in the provinces of Buenos Aires, Córdoba and Santa Fe; 19,233 shipments (14.2%) occurred between these 3 provinces; 14,303 shipments (10.6%) were between these 3 provinces and the other 20 provinces; 14,167 shipments (10.5%) were within each of the other 20 provinces and 4,758 shipments (3.5%) were between these 20 provinces (Fig 5 and Table 3).

Similar patterns could be seen when looking at the number of pigs shipped as opposed to the number of shipments (Table 3). However, when looking at net number of incoming/outgoing shipments between provinces, and disregarding internal provincial movements, there are some interesting relationships (Fig 5). The province with the largest net number of outgoing shipments is La Rioja, and the province with the largest net number of incoming shipments is La Pampa, with 381,598 (99.997% of outgoing pigs) of the pigs leaving La Rioja, going to La Pampa (89.5% of incoming pigs). Of these, 368,398 (96.5%) pigs were from a single movement pair between two units. A similar partnership can be seen between Córdoba and Santa Fe, with 284,005 pigs leaving Córdoba (59.2% of outgoing pigs) going to Santa Fe (59.7% of incoming pigs). Córdoba, La Rioja, Santa Fe, Buenos Aires and San Juan have all sent more than 100,000 pigs to other provinces from 2011 to 2016, making up 86% of between-provinces shipped pigs. Similarly, Santa Fe, La Pampa, Buenos Aires, Cordoba and Mendoza have all introduced more than 100,000 pigs, making up 88% of between-provinces shipped pigs (Fig 6).

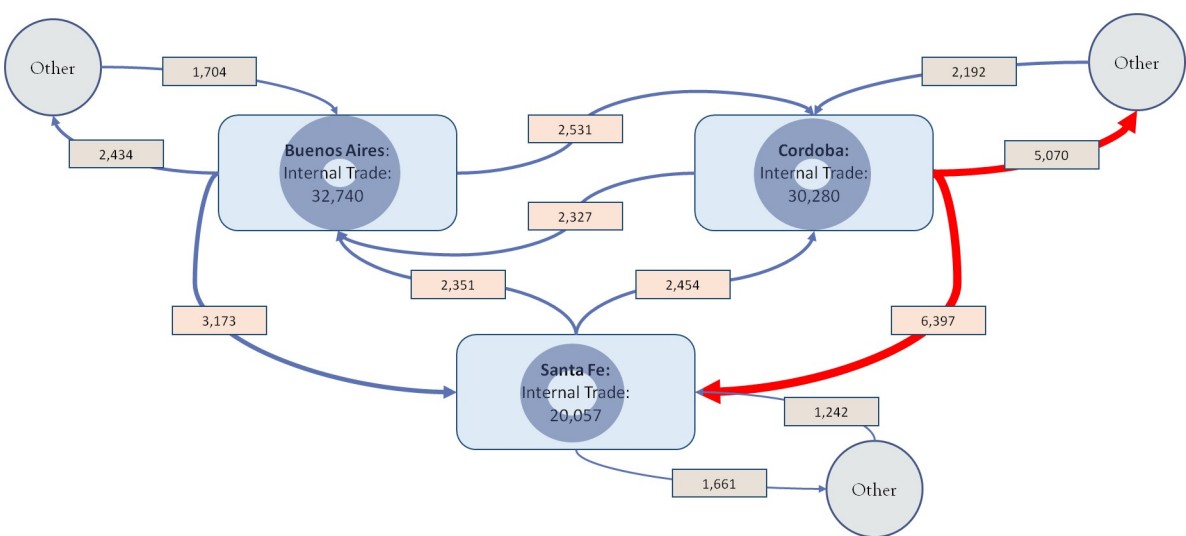

**Fig 5. Allocation of shipments involving the provinces of Buenos Aires, Cordoba and Santa Fe.** Edges thickness is proportional to the number of shipments, with the two most important trade relationships highlighted in red.

### 3.5 Network characteristics

As mentioned above, our unit/market-to-unit/market network had 135,538 edges, 17,809 nodes and 40,931 movement pairs. Of these pairs, 33,973 (83.0% of pairs) shipped between 1 and 99 pigs from 2011 and 2016 for a total of 670,660 pigs (11.3% of pigs), 6,237 (15.2% of pairs) shipped between 100 and 999 pigs for a total of 1,734,453 pigs (29.2% of pigs) and 717 (1.8% of pairs) pairs shipped between 1,000 and 70,000 pigs for a total of 2,602,224 pigs (43.8% of pigs). The remaining 4 movements pairs had shipped more than 160,000 pigs each for a total of 927,544 pigs (15.6% of pigs). Most movements were conducted over small distances with the median equal to 63 km. However, this was highly skewed with a mean of 142 km and a maximum distance of 3,286 km. Over the years both the median and mean distance slightly increased from 59 to 67 km and 136 to 152 km respectively (Table 4).

The monthly network graphs showed similar patterns, thus here we exemplify the 48th month of data, December 2014, which was one of the most legible graphs, to discuss some of the structures of the network (Fig 7). The first pattern we can notice is that a large portion of units and markets are connected, either directly or indirectly forming a major network community (Fig 7, number **1**). The rest of the units and markets for smaller communities, are disconnected from the main community (Fig 7, number **2**). In the main component, a very large blue node (market) surrounded by a multitude of direct connections can be seen, forming a dense star pattern (Fig 7, number **3**). We can also observe some small nodes (usually units) which connect communities that wouldn't otherwise be connected (Fig 7, number **4**). There are pairs of units that ship pigs to one another multiple times (Fig 7, number **5**). Though some major nodes exchanged pigs with other major nodes, some also exchanged pigs with a multitude of smaller units and, are only attached to another major node through a minor node (Fig 7, number **6**). Finally, we have movement pairs that are completely isolated from the rest of the network (Fig 7, number **7**).

Given the limitations of visualizing the network on a graph as it gets bigger (e.g. yearly network or full 6-year network), we also used a number metrics to help quantify some of the structures and network attributes noted above (Table 4). Overall the network is not very cohesive with a density that increased from 0.04 to 0.055% from 2011 to 2016. The majority of units (83,506; 85.6%) in the 2016 census did not send or receive a single shipment during the

**Table 3. Provincial distribution of shipments and number of shipped pigs in Argentina from 2011 to 2016.**

| | Internal shipments (within province) | Incoming shipments | Outgoing shipments | Internally shipped pigs (within province) | incoming pigs | outgoing pigs |
|---|---|---|---|---|---|---|
| **Main 3 provinces** | **83,077** | **24,371** | **28,398** | **3,848,169** | **828,330** | **837,858** |
| Buenos Aires | 32,740 | 6,382 | 8,138 | 1,243,754 | 192,302 | 169,528 |
| Cordoba | 30,280 | 7,177 | 13,794 | 1,119,648 | 160,396 | 480,045 |
| Santa Fe | 20,057 | 10,812 | 6,466 | 1,484,767 | 475,632 | 188,285 |
| **North East provinces** | **11,857** | **6,807** | **7,132** | **450,885** | **521,216** | **176,315** |
| Capital Federal | - | 186 | 185 | - | 963 | 419 |
| Chaco | 1,367 | 490 | 299 | 18,426 | 6,794 | 6,078 |
| Corrientes | 184 | 386 | 158 | 2,910 | 7,305 | 6,440 |
| Entre Rios | 2,290 | 1,481 | 1,263 | 160,381 | 24,783 | 30,966 |
| Formosa | 857 | 217 | 96 | 32,906 | 3,369 | 3,709 |
| La Pampa | 2,298 | 1,637 | 2,186 | 55,716 | 426,500 | 56,584 |
| Misiones | 2,365 | 268 | 69 | 141,915 | 6,732 | 694 |
| Salta | 837 | 435 | 575 | 18,795 | 10,786 | 18,912 |
| San Luis | 1,453 | 1,082 | 2,103 | 13,922 | 20,132 | 46,618 |
| Santiago del Estero | 206 | 625 | 198 | 5,914 | 13,852 | 5,895 |
| **Western and Southern provinces** | **2,310** | **7,116** | **2,764** | **91,088** | **195,193** | **530,566** |
| Catamarca | 51 | 535 | 174 | 1,042 | 12,480 | 6,428 |
| Chubut | 119 | 3 | 24 | 2,019 | 134 | 722 |
| Jujuy | 58 | 620 | 20 | 2,513 | 16,090 | 1,120 |
| La Rioja | 22 | 134 | 347 | 602 | 6,528 | 381,611 |
| Mendoza | 767 | 4,260 | 113 | 23,977 | 110,352 | 5,457 |
| Neuquen | 29 | 34 | 11 | 621 | 358 | 522 |
| Rio Negro | 759 | 189 | 134 | 45,341 | 2,870 | 1,086 |
| San Juan | 15 | 168 | 1,258 | 1,600 | 4,190 | 109,092 |
| Santa Cruz | 18 | 20 | 1 | 129 | 568 | 3 |
| Tierra del Fuego | 3 | 3 | - | 246 | 148 | 0 |
| Tucuman | 469 | 1,150 | 682 | 12,998 | 41,475 | 24,525 |
| **Total** | **97,244** | **38,294** | **38,294** | **4,390,142** | **1,544,739** | **1,544,739** |

period 2011–2016. A narrow majority of units that did participate in movements only moved once or twice over the whole period (9,146; 50.4%), with another 5,042 units (27.8%) participating in 3 to 10 shipments. When looking at indegree and outdegree, we observe that over the years the median is 1, with the mean varying between 3 and 3.5. The 95th percentiles are also low, around 7 to 11. However, we can see that the maximums are well above 1,000 for indegree, and around 700 for outdegree each year. Similarly, when looking at betweenness we see a relatively high mean but the median being 0, with some very extreme maximums, between 600,000 and 3,000,000 depending on the year. At the month level, these metrics showed marked seasonal patterns with degree and density peaking in September and October (Fig 8) and betweenness showing even stronger peaks in August, September and October (Fig 9).

When looking at strong and weak components, we can see a very large number of strong components (between 5,500 and 6,900 depending on the year), and a small number of weak components (between 431 and 568 depending on the year) (Table 4). In the weak components we have one very large component which includes 79 to 83% of all nodes for any given year. All other weak components are much smaller (mean size 2). Strong components are much smaller on average with a mean and median of 1 and yearly maximums ranging from 270 to 564.

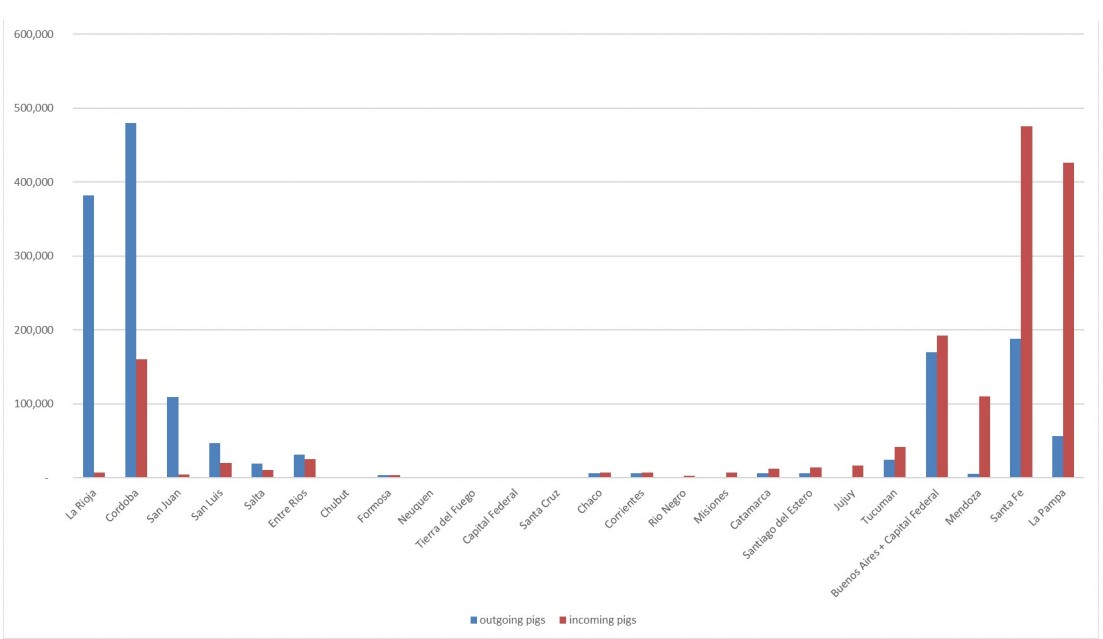

**Fig 6. Distribution of inter-provincial traded pigs in Argentina between 2011 and 2016.**

### 3.6 The role of markets

During the total period of study, 11,394 shipments, involving 251,449 pigs arrived from units to markets, 11,812 shipments involving 183,465 pigs went from markets to units and 1,201 shipments and 59,477 pigs going directly from market to slaughter. Thus, markets represent 1.8% of nodes but are involved in 17.4% of shipments and 7.3% of pigs moved. A total of 6,420 movement pairs (15.6% of all pairs) involved markets. Shipments involving markets were much smaller, being on average of 19 pigs, with a median of 8 and a maximum of 487. As mentioned earlier, one particular market stands out, having much higher values for each of the network metrics, compared to any other nodes in the network. This node is mapped with the other 350 markets involved in the network, in Fig 10. We can see that a few markets form star patterns (Fig 10), reflecting that they are connected to multiple units, and so hold strategic positions. The important role of markets is confirmed by the metrics, with markets having higher mean and median in and outdegree (mean of 7.5 and median of 1 for the full network compared to, 18.4 and 6 for market outdegree and 14.5 and 2 for market indegree). The same was observed for betweenness (mean of 318,732 for markets compared to 20,673 for the full network). When building the network without the markets (Table 5), though network density is only slightly different, with markets representing a small proportion of nodes, it is noticeable that eigen centrality for the network is quite lower. Mean yearly betweenness also dramatically decreases from 708 to 3,895 in the full network compared to 22 to 238 in the network without markets. Finally, we can note that largest weak component now only includes 74 to 77% of nodes compared to 79 to 83% to the full network. The new network has 16,145 nodes, 2,015 less compared to the full network. This means, that after accounting for 351 markets, 1,664 units have been excluded as they were moving pigs exclusively with markets.

When looking at monthly metrics, monthly density and degree follow similar patterns as previously observed, with slightly lower values. A moderate peak is still visible around October (Fig 11). However, the seasonal patterns observed in the full network for betweenness disappear with no obvious yearly peak in August to October (Fig 12). The seasonality in betweenness was driven by two specific markets mostly operating in August, October and November.

**Table 4. Network centrality values and characteristics for the yearly networks from 2011 to 2016 and the complete network (whole time period).**

| | 2011 | 2012 | 2013 | 2014 | 2015 | 2016 | Total |
|---|---|---|---|---|---|---|---|
| **Network attributes** | | | | | | | |
| Number of farms & markets (nodes) | 5,852 | 8,110 | 7,603 | 6,807 | 6,784 | 6,323 | 18,160 |
| Number of shipments (edges) | 14,897 | 25,655 | 25,826 | 23,695 | 23,355 | 22,110 | 135,538 |
| Number of pigs shipped | 509,965 | 962,006 | 1,015,450 | 1,091,699 | 1,136,035 | 1,219,726 | 5,934,881 |
| **Euclidean distance (edge length), km** | | | | | | | |
| Median | 66.5 | 59.4 | 60.6 | 62.3 | 65.7 | 67.9 | 63.3 |
| Mean | 142.8 | 136.2 | 135.5 | 141.8 | 147.1 | 152.2 | 142.3 |
| 95th percentile | 520.3 | 527.0 | 520.6 | 540.2 | 544.2 | 553.1 | 539.4 |
| Maximum | 1,404 | 2,519 | 2,872 | 3,286 | 1,591 | 1,278 | 3,286 |
| **Shipment size** | | | | | | | |
| Median | 15 | 17 | 18 | 20 | 20 | 20 | 20 |
| Mean | 34 | 38 | 39 | 46 | 49 | 55 | 44 |
| 95th percentile | 113 | 120 | 125 | 150 | 160 | 200 | 141 |
| Maximum | 7,362 | 7,819 | 3,000 | 10,727 | 14,443 | 10,370 | 14,443 |
| **Indegree** | | | | | | | |
| Median | 1 | 1 | 1 | 1 | 1 | 1 | 1 |
| Mean | 2.55 | 3.16 | 3.4 | 3.48 | 3.44 | 3.50 | 7.46 |
| 95th percentile | 7 | 8 | 10 | 10 | 11 | 11 | 21 |
| Maximum | 1,337 | 1,361 | 1,423 | 1,252 | 1,354 | 1,239 | 7,966 |
| **Outdegree** | | | | | | | |
| Median | 1 | 1 | 1 | 1 | 1 | 1 | 1 |
| Mean | 2.55 | 3.16 | 3.4 | 3.48 | 3.44 | 3.50 | 7.46 |
| 95th percentile | 9 | 12 | 12 | 13 | 13 | 13 | 29 |
| Maximum | 777 | 658 | 730 | 744 | 661 | 708 | 4,278 |
| **Betweenness** | | | | | | | |
| Median | 0 | 0 | 0 | 0 | 0 | 0 | 0 |
| Mean | 771 | 1,723 | 3,895 | 886 | 1,398 | 708 | 20,673 |
| 95th percentile | 538 | 1,754 | 2,571 | 981 | 1,259 | 918 | 28,143 |
| Maximum | 687,196 | 1,238,883 | 2,999,446 | 726,797 | 1,172,503 | 601,057 | 30,558,417 |
| **Eigen Centrality** | -57.12 | 41.86 | 48.45 | 41.38 | 45.93 | 51.93 | 260.62 |
| **Network density (%)** | 0.0435 | 0.0390 | 0.0447 | 0.0511 | 0.0508 | 0.0553 | 0.0411 |
| **Strong components** | | | | | | | |
| Number | 5,486 | 7,499 | 6,911 | 6,358 | 6,303 | 5,882 | 14,314 |
| Largest component | 287 | 464 | 564 | 270 | 344 | 285 | 3,546 |
| Median size | 1 | 1 | 1 | 1 | 1 | 1 | 1 |
| Mean size | 1.07 | 1.08 | 1.10 | 1.07 | 1.08 | 1.08 | 1.27 |
| **Weak components** | | | | | | | |
| Number | 431 | 568 | 533 | 497 | 493 | 485 | 923 |
| Largest component (% nodes) | 4,843 (82.8) | 6,684 (82.4) | 6,199 (81.5) | 5,444 (80.0) | 5,468 (80.6) | 4,994 (79.0) | 16,000 (88.1) |
| Median size | 2 | 2 | 2 | 2 | 2 | 2 | 2 |
| Mean size | 13.58 | 14.28 | 14.26 | 13.7 | 13.76 | 13.04 | 19.67 |

## 3.7 Factor analysis for mixed data

Due to a number of observations having some missing data in the variables of interest, analysis was conducted on 14,099 complete observations out of the 18,160 nodes which participated in the network. A first model was constructed using FAMD which included all the variables originally selected. FAMD plots variables in a multi-dimensional space based on how they interact,

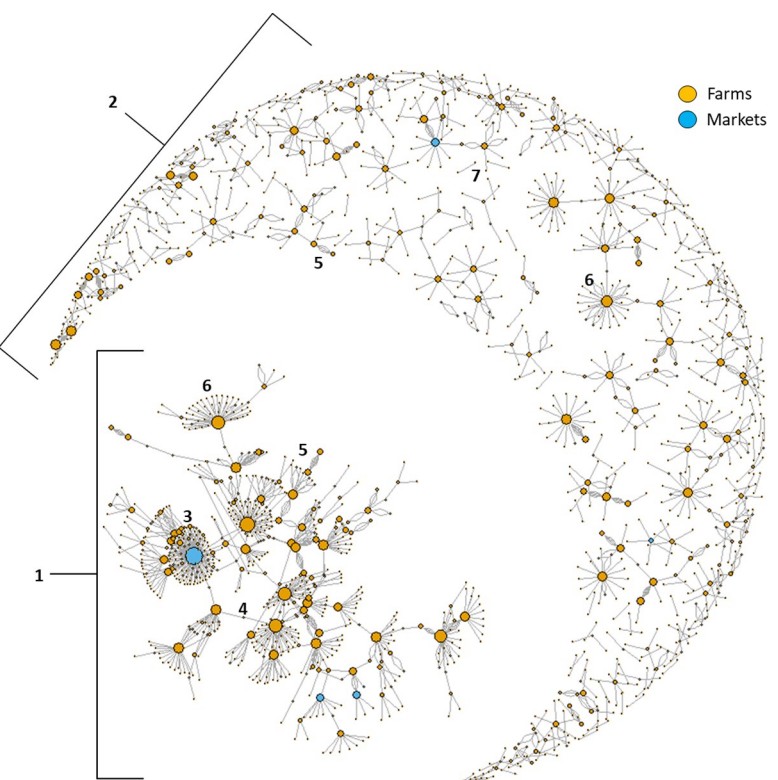

**Fig 7. Network graph for December 2014.** Node size represents the log value for node betweeness.

with coordinate values ranging from -1 to 1 in each dimension. The number of dimensions is dependent on the number of variables, and the number of categories within categorical variables [24]. Typically, results are interpreted in the first 2 or 3 dimensions as these explain most of the data variation. Values close to 0 reflect variables with low discriminatory power, thus variables which do not divide observations in distinct groups. Coordinate values closer to -1 or 1 have high discriminatory power. For this study, variables which did not reach coordinate values of 0.2 in either dimension 1 or 2 were excluded as active variables and only included as

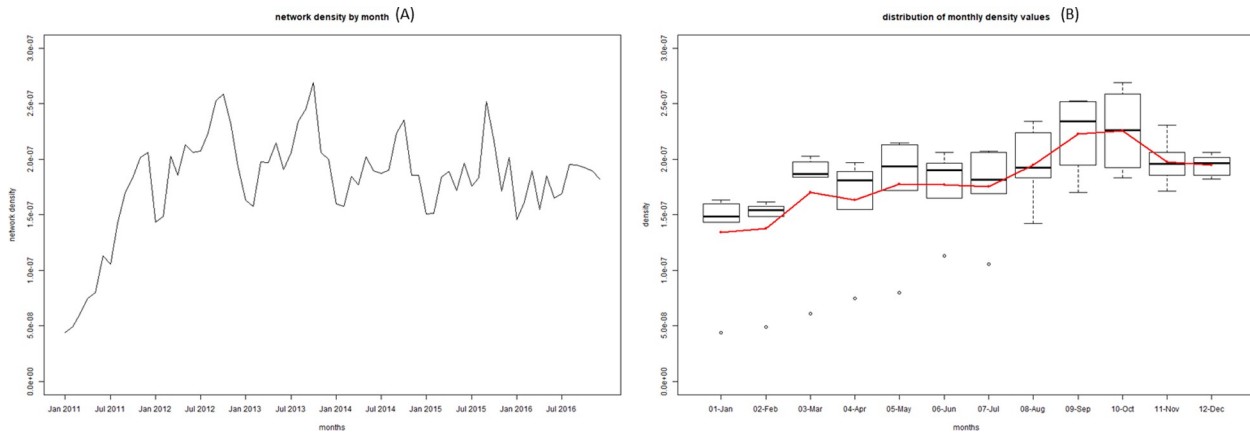

**Fig 8.** Time series of the monthly density values from 2011 to 2016 (A) and boxplot of the density values aggregated by month from 2011 to 2016 (B).

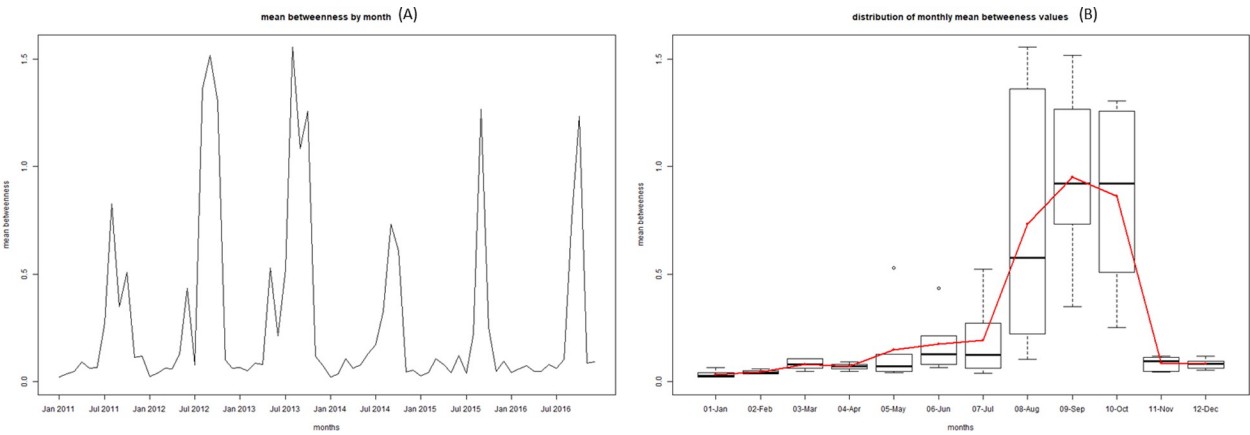

**Fig 9.** Time series of the monthly mean betweenness value from 2011 to 2016 (A) and boxplot of the betweenness values aggregated by month from 2011 to 2016 (B).

passive supplementary variables, meaning that these variables were not discriminatory enough to help divide nodes into groups. These non-discriminatory variables included unit area, non-swine livestock population, poultry population and distance covered by incoming and outgoing shipments. Province was also excluded from the model as the large number of categories created a large number of dimensions which explained very little of the variance and made it impossible to define clear groupings. The final model was studied in the first two dimensions, which accounted for 27.5% and 23.6% of the total variance respectively. As these accounted for more than half of the variance, and dimension 3 dropped to 14.1% of variance, results were interpreted using graphical representations in the first two dimensions (Fig 13). The three continuous variables with the highest contribution (i.e., the most discriminatory power) were indegree, outdegree and betweenness in both dimension 1 and 2 (Table 6). The active variables all had positive coordinates in both dimensions, thus trending towards discriminating nodes with a combination of higher values. This means that high values of one variable tends to combine with high values of the other variables as well within observations. These variables are plotted in Fig 13.

Hierarchical clustering suggested the optimal number of four clusters (Fig 14). These could be defined as cluster 1 comprised of small units (low pig population) with a combination of low network metrics (betweenness and in- and out-degree), low shipment sizes; cluster 2 comprised of large units with a combination of low betweenness but high in- and out-degree and large shipment sizes; cluster 3 comprised of markets (no pig population) with a combination of high betweenness but low in- and ou-degree and small shipment size; and cluster 4 comprised of a single node, the major outlying market mentioned previously with extremely high values for betweenes and in- and ou out-degree but low shipment sizes. Detailed results of the values within each cluster are presented in Table 7.

## 4. Discussion

Swine farming in Argentina may not represent the bulk of meat production compared to cattle or poultry in the country. However, it is still an important sector both at the industrial and community level and has a substantial potential to grow and expand, particularly now that other traditional pig producing regions such as China or Europe have been affected by ASF. Swine farming in Argentina is divided into two main groups, an industrial production sector, and small-scale and backyard farming. Thus, every value relating to pig numbers or

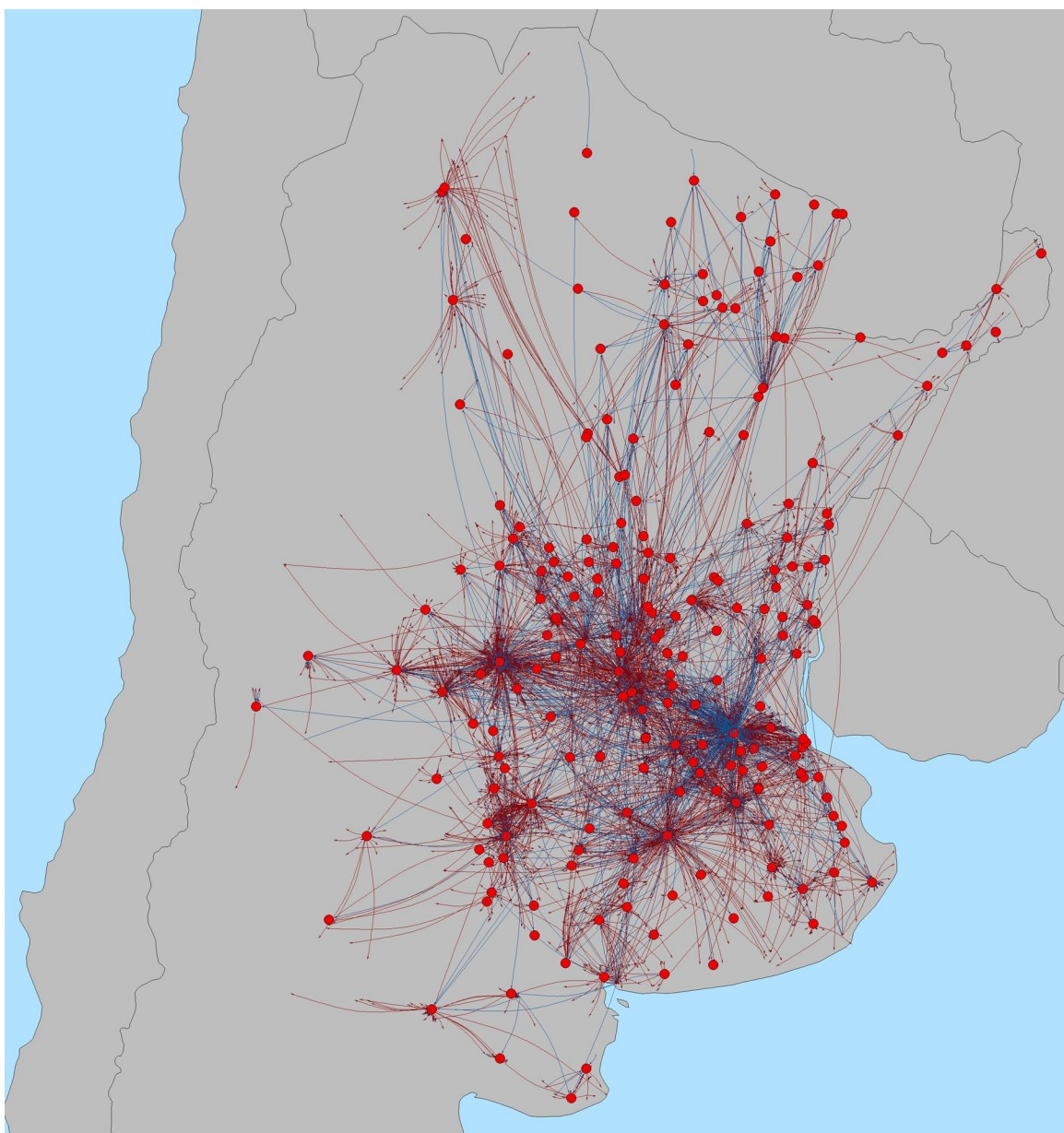

**Fig 10. Map of shipments going to or coming from markets, from 2011 to 2016.** Only market nodes are shown. Red lines represent shipments coming from markets, and blue lines, coming from farms.

movements are highly skewed with a large majority of small holdings and a few very large holdings that ship large numbers of animals. The presence of other livestock species was also indicative of the difference between large and small swine units, with large swine units having little to no other livestock on the premise, and most other livestock being on units with small or medium sized swine herds. The industrial-backyard dichotomy was evident when mapping the units, with swine density not fully matching unit density. The open plains west of Buenos Aires contain the so-called industrial swine belt of Argentina with high densities of both pig units and number of pigs. The more densely forested Northeast is home to a large amount of backyard farmers, which explains the very high density of units with a low density of pigs.

**Table 5. Network centrality values and characteristics for the yearly sub-networks from 2011 to 2016 and the full sub-network (whole study period) after removing markets.**

| | 2011 | 2012 | 2013 | 2014 | 2015 | 2016 | Total |
|---|---|---|---|---|---|---|---|
| **Network attributes** | | | | | | | |
| Number of farms & markets (nodes) | 4,928 | 7,134 | 6,737 | 6,148 | 6,113 | 5,662 | 16,145 |
| Number of shipments (edges) | 11,002 | 21,097 | 21,443 | 20,248 | 19,724 | 18,820 | 112,334 |
| Number of pigs shipped | 442,382 | 889,119 | 939,700 | 1,022,790 | 1,063,184 | 1,142,792 | 5,499,967 |
| **Euclidean distance (edge length), km** | | | | | | | |
| Median | 80.9 | 64.8 | 64.1 | 65.1 | 69.3 | 70.3 | 67.9 |
| Mean | 164.4 | 146.6 | 143.6 | 150.2 | 157.7 | 162.2 | 153.0 |
| 95th percentile | 549.6 | 545.1 | 540.6 | 551.9 | 570.2 | 578.2 | 554.7 |
| Maximum | 1,404 | 2,519 | 2,872 | 3,286 | 1,591 | 1,278 | 3286 |
| **Shipment size** | | | | | | | |
| Median | 20 | 20 | 20 | 22 | 20 | 23 | 20 |
| Mean | 40 | 42 | 44 | 51 | 54 | 61 | 49 |
| 95th percentile | 130 | 126 | 140 | 167 | 185 | 200 | 156 |
| Maximum | 7,362 | 7,819 | 3,000 | 10,727 | 14,443 | 10,370 | 14,443 |
| **Indegree** | | | | | | | |
| Median | 1 | 1 | 1 | 1 | 1 | 1 | 1 |
| Mean | 2.23 | 2.96 | 3.18 | 3.29 | 3.23 | 3.32 | 6.96 |
| 95th percentile | 6 | 7 | 9 | 10 | 10 | 11 | 20 |
| Maximum | 351 | 1,293 | 967 | 837 | 733 | 511 | 4,547 |
| **Outdegree** | | | | | | | |
| Median | 1 | 1 | 1 | 1 | 1 | 1 | 1 |
| Mean | 2.23 | 2.96 | 3.18 | 3.29 | 3.23 | 3.32 | 6.96 |
| 95th percentile | 8 | 11 | 12 | 12 | 12 | 12 | 27 |
| Maximum | 401 | 388 | 368 | 400 | 417 | 377 | 1,955 |
| **Betweenness** | | | | | | | |
| Median | 0 | 0 | 0 | 0 | 0 | 0 | 0 |
| Mean | 22 | 238 | 109 | 86 | 85 | 66 | 17,832 |
| 95th percentile | 20 | 213 | 84 | 56 | 79 | 54 | 18,879 |
| Maximum | 8,514 | 241,007 | 46,307 | 40,735 | 27,899 | 22,443 | 17,995,571 |
| **Eigen Centrality** | 17.15 | 21.91 | 21.63 | 23.35 | 32.37 | 35.77 | 90.28 |
| **Network density (%)** | 0.0453 | 0.0415 | 0.0473 | 0.0536 | 0.0528 | 0.0587 | 0.0431 |
| **Strong components** | | | | | | | |
| Number | 4,832 | 6,918 | 6,516 | 5,949 | 5,919 | 5,483 | 13,446 |
| Largest component | 8 | 47 | 47 | 51 | 45 | 29 | 2,278 |
| Mean size | 1 | 1 | 1 | 1 | 1 | 1 | 1 |
| Median size | 1.02 | 1.03 | 1.03 | 1.03 | 1.03 | 1.03 | 1.20 |
| **Weak components** | | | | | | | |
| Number | 483 | 644 | 611 | 539 | 537 | 531 | 1,049 |
| Largest component (% nodes) | 3,801 (77.1) | 5,484 (76.9) | 5,134 (76.2) | 4,570 (74.3) | 4,693 (76.8) | 4,229 (74.7) | 13,660 (84.6) |
| Mean size | 2 | 2 | 2 | 2 | 2 | 2 | 2 |
| Median size | 10.20 | 11.08 | 11.03 | 11.41 | 11.38 | 10.66 | 15.39 |

Both these areas are also where human population density is the highest [25]. Finally, the less densely populated mountainous regions in the West, bordering Chile, and desert regions in the South is home to fewer small backyard holdings. However large holdings still exist in limited numbers across the country, as noted by the skewness in unit size for each province.

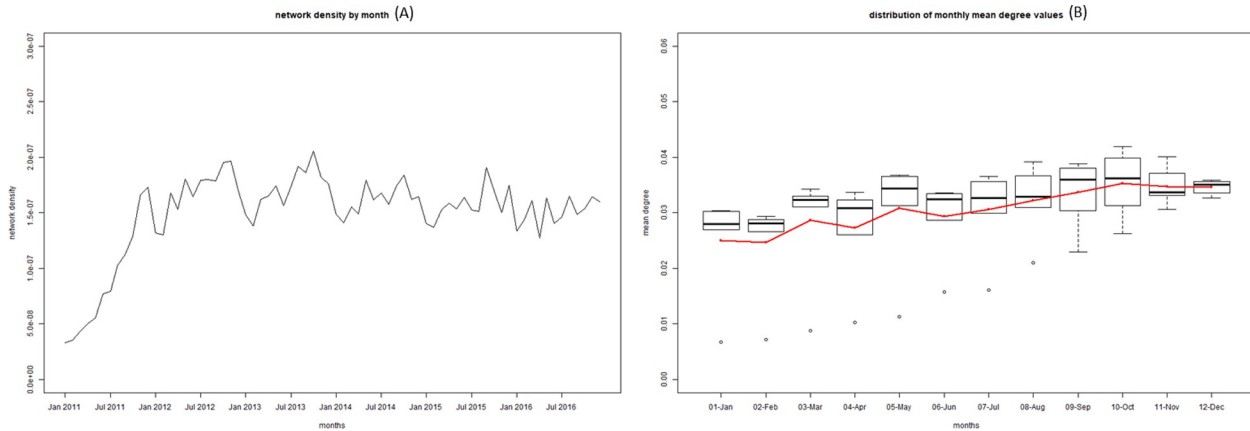

**Fig 11.** Time series of the monthly density value from 2011 to 2016 in the network without markets (A) and boxplot of the betweenness values aggregated by month from 2011 to 2016 in the network without markets (B). These graph are on the same scale as Fig 8.

The density of recorded shipments, ingoing or outgoing, was also very focalized around certain specific points in the industrial swine belt and this is in part because a few very large holdings and markets were involved in very large portion of shipments. Some of these holdings have near exclusive partnerships with other large holding, repeatedly sending large shipments in one direction. Though the data did not have enough information to confirm this, we suspect that this represents large breading farms sending many young pigs to large fattening or finishing farms, as is relatively common in industrial swine farming worldwide [5,8].

Seasonal trends were less clear to interpret. It is likely that to a certain extent movements reflect seasonal patterns in swine farming as has been shown in the cattle industry [11, 26]. During the summer, pig fattening is less efficient and animals are relatively small when they are shipped out in December-January to make way for new incoming piglets that will start the fattening process. Given the smaller size of shipped animal, more animals are shipped on fewer trucks explaining that there are fewer shipments during those months but with a larger average number of pigs per shipment. However, the increase in the number of shipments in October-November, might also reflect the industry getting ready for increased meat consumption that always occurs during the spring and summer. Movements to and from markets showed a major increase in September and October which might also reflect the period of

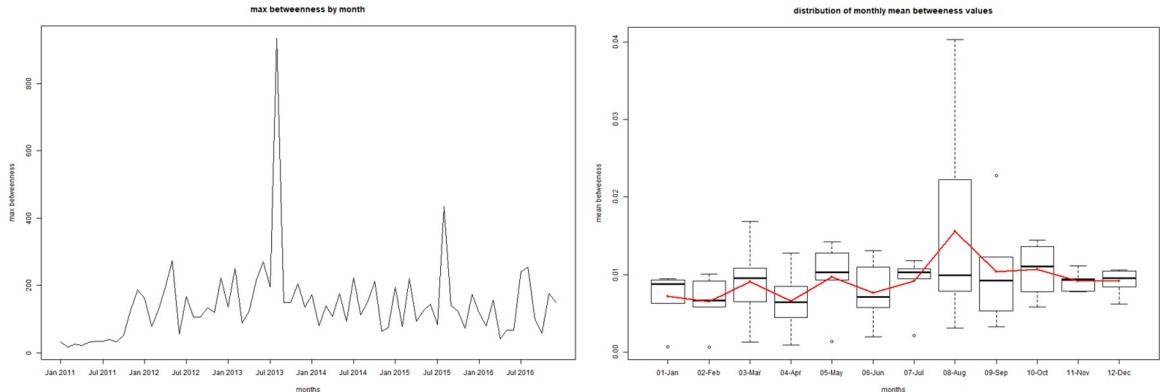

**Fig 12.** Time series of the monthly mean betweenness value from 2011 to 2016 in the network without markets (A) and boxplot of the betweenness values aggregated by month from 2011 to 2016 in the network without markets (B).

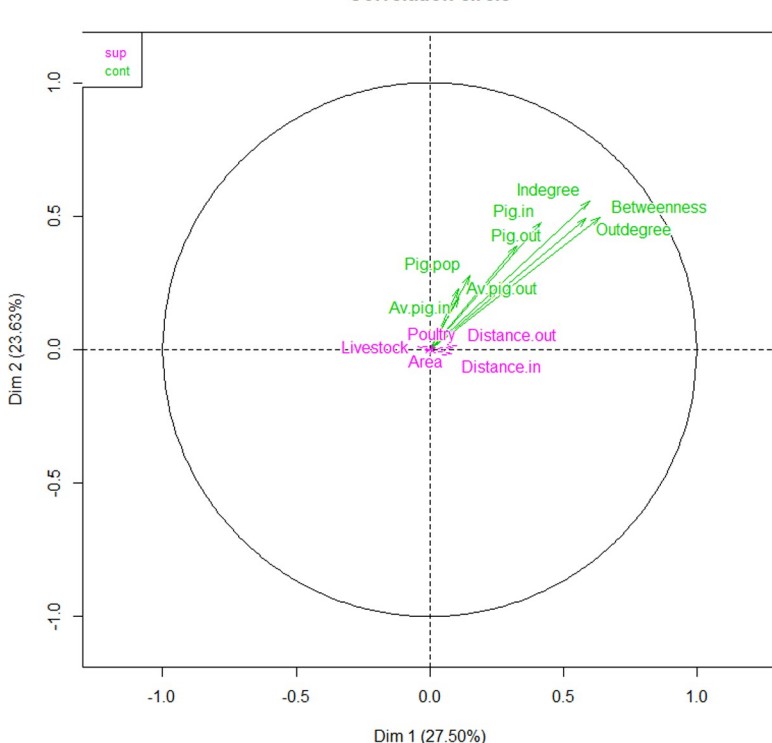

**Fig 13. Correlation circle for continuous variables included in the final factor analysis for mixed data model (FAMD) along dimension 1 and 2.** Variables in green are active and in purple are supplementary. All active variables have positive coordinate values in both dimension 1 and 2. The variable names relate to: indegree, outdegree, betweenness, Pig.in = total number of incoming pigs in a given unit, Pig.out = total number of outgoing pigs, Pig. pop = pig population, Av.pig.out = average size of outgoing shipment Av.pig.in = average size of incoming shipment, Polutry = poultry population, Livestock = livestock population, Distance.out = mean distance of outgoing shipment, Distance.in = mean distance of incoming shipment, Area = area of unit.

activity of certain markets, thus affecting the overall picture as well. This is confirmed by the important peaks in mean betweenness during those same months which were mainly driven by markets indirectly connecting multiple units that otherwise would not have been in contact. The fact that shipments through markets drive the peaks in the number of shipments in September and October might also be an explanation as to why these months had on average the smallest shipment sizes, as shipment through markets were smaller than shipments between units only. The repeating patterns over the years 2011 to 2016 showed that shipments follow a stable network that can be predicted with some reliability over future years. The observed trend of decreasing shipments over the years whilst the number of shipped pigs increased might reflects financial strain due to the economic crisis, with farmers aiming to cut cost on shipments by shipping more pigs in a shipment.

Network graphs can provide valuable insights about network structure and can help to identify key nodes where surveillance and outreach should be focused. For example, in the December 2014 graph (Fig 7) we noticed an important market located just west of Buenos Aires which consistently plays on central role between a large number of units, connecting with both major and minor nodes in the network. This same market drives the seasonal variation in betweenness with peaks from August to October, whilst remaining active the rest of the year also. As noted, we distinguished several patterns in this graph which help to illustrate the fact that the number of animals and shipments a node ships is not directly associated with the

**Table 6. Results of factor analysis for mixed data (FAMD).** Coordinates represent the mean location of a variable along the 2 dimensions under study, and contribution represents the discriminatory power of the given variable in dimension 1 or 2. (Supplementary categorical variable of province not shown for clarity, due to the large number of provinces).

| | Dimension 1 | | Dimension 2 | |
|---|---|---|---|---|
| | Coordinates | Contribution | Coordinates | Contribution |
| **Active Continuous Variables** | | | | |
| Betweenness | 0.636 | 13.71 | 0.497 | 9.73 |
| Indegree | 0.598 | 12.11 | 0.556 | 12.21 |
| Outdegree | 0.585 | 11.61 | 0.490 | 9.45 |
| Number of incoming pigs | 0.417 | 5.90 | 0.478 | 9.00 |
| Number of outgoing pigs | 0.324 | 3.56 | 0.388 | 5.92 |
| Pig population | 0.149 | 0.76 | 0.278 | 3.04 |
| Average incoming shipment size | 0.109 | 0.40 | 0.195 | 1.50 |
| Average outgoing shipment size | 0.107 | 0.39 | 0.230 | 2.09 |
| **Active Categorical Variable categories** | | | | |
| Node type: Market | 5.927 | 50.70 | -4.865 | 46.26 |
| Node type: Farm | -0.101 | 0.87 | 0.083 | 0.79 |
| **Supplementary Continuous Variables** | | | | |
| Area | -0.016 | NA | 0.001 | NA |
| Non-pig livestock population | -0.046 | NA | 0.013 | NA |
| Poultry population | 0.007 | NA | 0.018 | NA |
| Average distance of incoming shipments | 0.082 | NA | -0.017 | NA |
| Average distance of outgoing shipments | 0.102 | NA | 0.017 | NA |

importance of a node in the network in terms of connecting units and markets and potentially contributing to a high potential of disease spread. Small nodes which only connect between two other nodes and only have 2 shipments, might not appear important in terms of number of shipments and pigs shipped, without visualizing the network and network metrics, but can be located in strategic positions in the network and serve as a bridge by indirectly connecting other groups of nodes or two highly connected communities that otherwise will be not connected. Conversely nodes that send repeated number of shipments to a single other node might not necessarily have an important role in the network in terms of disease spread, despite the large number of shipped animals, as they only contact one or a few partners and can be in an isolated circuit. The repeated nature of shipments between two nodes was also exemplified above with only 4 movement pairs exchanging nearly 16% of swine shipped. Repeated exchanges often occur when there is a specific partnership between a large breading unit and a large fattening unit, with the piglets going from one to the other at regular intervals, without going further into the farm network. Thus, a large portion of shipments could be considered to have relatively low importance for surveillance and control needs. Though some major nodes followed this repeated pair patterns with other major nodes, some send shipments to multiple smaller nodes, and were only attached indirectly to another major node. Here we have much higher spreading risks as these units hold more central role in the various communities. Finally, we noted some isolated pairs outside of the main network which are likely to be backyard farmers that might exchange a few pigs or a boar, with a neighbor without ever contributing to the network as whole. These units can also be seen on the periphery of the communities and a large number of them only participated in pig movements once in the whole 6 years of our time period. However, there were two major limitation in assessing indirect connections between nodes. Firstly, the networks presented in this paper were all static, even if observed at different time-scales. This creates an issue, where, for example: in a monthly

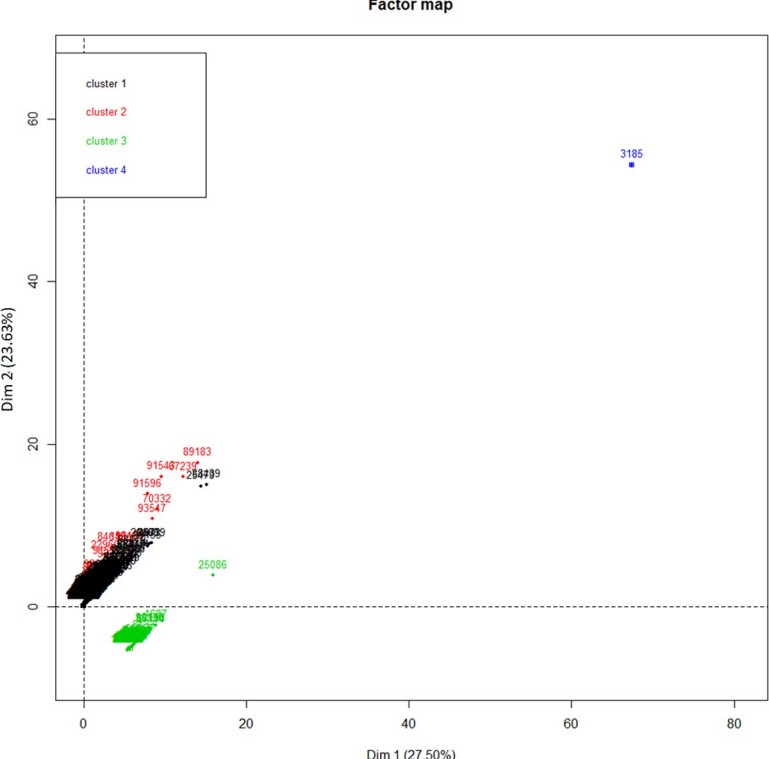

**Fig 14. Location of nodes in dimensions 1 and 2 following coordinates obtained from factor analysis of mixed data (FAMD) with color coding from hierarchical clustering.** Cluster 1 represents small and backyard productive units (low degree, betweenness, pig population and shipment size), cluster 2 represents large and industrial farms (high degree, pig population and shipment size but low betweenness), Cluster 3 represents markets (high betweenness but low degree and shipment size, and no pig population) and cluster 4 is a single outlying market with extremely high values for betweenness, degree but small shipment size and no pig population.

network, two farms A and C are connected via two separate shipments through farm B. These two shipments might be distant in time by one day or by as much as thirty, and this time-lapse might make the indirect connection note-worthy or irrelevant in terms of disease transmission

**Table 7. Characteristics of the four clusters defined by hierarchical clustering based on the active variables selected from factor analysis for mixed data (FAMD).**

| Variables | Cluster 1 | Cluster 2 | Cluster 3 | Cluster 4 |
|---|---|---|---|---|
| | Farms with lower trade metrics | Farms with high trade metrics | Markets with lower trade metrics | Market with high trade metrics |
| Number of nodes in cluster | 13,845 | 17 | 236 | 1 |
| Mean Betweenness | 19,617 | 12,754 | 190,598 | 30,558,417 |
| Mean Indegree | 7.8 | 117.5 | 14.5 | 7,966 |
| Mean Ourdegree | 8.1 | 132.7 | 31.9 | 4,278 |
| Mean Number of incoming pigs | 299 | 62,160 | 214 | 200,857 |
| Mean Number of outgoing pigs | 298 | 64,078 | 144 | 149,550 |
| Mean Pig population | 226 | 22,246 | 0 | 0 |
| Average incoming shipment size | 14 | 1,630 | 12 | 25 |
| Average outgoing shipment size | 19 | 1,458 | 5 | 35 |

from A to C. This problem would be exacerbated with larger time steps. It would thus be useful to use a dynamic network structure to better capture the risk of transmission through indirect connections, considering the incubation period, latent period and other temporal characteristics of any specific disease under study. Secondly, the lack of direct animal tracing meant it wasn't possible to assess which shipments continued directly from a node A to C via B. This is especially true with markets where numerous shipments come in and exit at any given time, with no resident population. Thus, if farm A, sends a shipment to market B, and shortly after, farm C receives a shipment from market B, there is no way to know if this shipment contains the same pigs sent by farm A, or other pigs that were at the market at the same time, and which might or might not have been in close contact to pigs from farm A. This is certainly something that could be added if individual pig identification expands in Argentina and that information becomes available for analysis in the future.

It is also interesting to discuss and compare the metrics we obtained with those described in previous studies. Overall, monthly network density values varying between $1.5*10^{-7}$ and $2.5*10^{-7}$ (Fig 8A) reflected a very loose and disjointed network. These values are much lower than that seen in countries with a much larger swine industrial sector but little backyard swine production such as the United States of America, Canada and Germany, [5, 6, 8], with values ranging $3*10^{-3}$ and $8*10^{-3}$. This can be explained by the very large number of nodes that never engaged in shipping or receiving pigs. It is likely that restricting the network to large industrial farm we would reach density values similar to that shown in the examples above.

Centrality metrics reflected generally the patterns observed in the graph. The very low mean yearly indegree and outdegree confirmed the fact that the vast majority of units participated very little in pig movements over the 6 years under study. However, the very large maximum indegree and outdegree, between 650 and 1,450, each year all relate to the market mentioned above, with the next most important nodes being a few units with several dozens to hundreds of shipments, with yearly maximums ranging from 350 to 1,300. We see here the highly skewed nature of shipments with a few nodes concentrating a large portion of shipments. This pattern was also noted in the Argentinian bovine industry [11, 26]. The mean and median betweenness of 0 each year relates to the fact that most units are peripheral to the network contributing one shipment in one direction or the other without ever connecting indirectly two or more nodes. Once again, the extreme values are from the major market near Buenos Aires. Interestingly, some units with high degree values did not also have high betweenness values. Nodes with high degree values and low betweenness are linked to the pairing partnership discussed above, between a specific breading unit sending multiple shipments to a specific fattening unit, without connecting much with other nodes. In the context of infectious disease, though in and outdegree give useful information in regards to the intensity of movements, betweenness is most interesting in terms of finding nodes with strategic locations in the network and where surveillance would be the most useful. In this sense we can see that markets have a role in the swine movement network in Argentina disproportionate to the number of shipments and pigs that actually go through them. Not only were degree values on average higher for markets than for productive units but taking markets out of the network dramatically reduced the monthly betweenness of the network, presumably breaking up the network into smaller less connected components. Moreover, removing markets from the network also drastically reduced the overall eigen centrality, indicating the role of markets in indirectly connecting multiple communities of nodes together. Thus, markets play key roles in indirectly connecting units that would not have been connected otherwise. Once again, we have to take into account the limitations of static monthly networks and the lack of continued shipment tracing in trying to assess the value of an indirect connection for disease transmission. The fact that monthly centrality values followed repeated patterns over the years,

particularly betweenness, is interesting in terms of being able to predict periods of strategic importance with the months of August, September and October being of crucial importance. Thus, by combining information about time and place, we can select specific nodes, productive units and markets, that would be of crucial importance for a control campaign or outbreak management at specific times of the year. Eigen centrality is another way we can distinguish nodes with high connectivity, with strategic roles in the network.

Community algorithms also reflected what we observed in the graph with one large community containing most nodes in any given month, which included multiple strong components connected indirectly by a few nodes, and a multiplicity of small independent communities on the periphery. However, the largest community still contained a smaller proportion of nodes compared to examples in more industrial production systems in the US and Germany [5, 8] where more than 90% of nodes were contained in the largest community. This could be explained by the fairly large proportion of backyard producers that do not exchange pigs with more industrial facilities. However, the proportion of units involved in the large community in Argentina remained much larger compared to examples in Canada, France, Italy and Spain, which also have industrial swine production systems [6, 7]. Given that multiple factors might divide a single country's sector in multiple communities, such as type of farming, but also presence of industrial groupings or partnerships, geography and natural barriers or the role of markets, it is difficult to draw direct comparison between systems without looking more closely into a more detailed layout of the communities. Looking at geographical clustering of communities [7], would be a next step in characterizing the Network structure in Argentina. Seeing that most shipments remain within a given province does provide some evidence of potential clustering of communities within provinces. The large amount of small strong components reflect that most shipping partnerships are in pairs, with a few nodes branching out into star patterns, connecting directly with multiple other nodes. This community structure was repeated across years and months.

Furthermore, factor analysis re-enforced the notion that a small group of large industrial farms play a disproportionate role in sending and receiving swine shipments in terms of volume and could help identify the most crucial of these. The vast majority of farms being small scale enterprises that do not participate much in pig movements, if at all. However, in it is interesting to note that when comparing values between clusters 1 and 2, though cluster 2 had on average values much superior to cluster 1 in almost every variable included in the model, this was not the case for an important exception, betweenness. In this regard small units in cluster 1 had an average betweenness slightly higher than cluster 2 and much lower than the market clusters. This relates to the point mentioned above about smaller nodes with low degree and high betweenness which are likely small holdings moving pigs at "random" as opposed to large holdings sending a large number of repeated shipments to a select few other units, and thus not being necessarily as important in the network as the large number of shipments suggests. It is also notable when looking at Fig 14, that though cluster 3 and 4 are distinctly different from each other and cluster 1 and 2, there seem to be some level of overlap between cluster 1 and 2, despite the large differences in mean values. This suggests, that though we can divide units into two broad types, there is no clear limit between these, and a number of units have more intermediate values. Here again, markets appear to have a much more important role in pig movements even when removing the one major outlier which formed its own spate cluster.

## 5. Conclusion

The characterization of the network structure of swine movements in Argentina provides useful information to build targeted and cost-effective surveillance and control system in an area

of the world where the swine industry has been little studied to date. Such network structures can be adapted to create dynamic disease transmission models for multiple agents to test the impact of risk-based surveillance and intervention to help eradicate endemic diseases such Aujesky's disease and to predict the impact of the potential introduction of new pathogens such as the PRRS, ASF and CSF viruses. However, to fully assess the risk and impact of introduction of pathogens for which Argentina is currently free, data about pig imports would be crucial. This would allow the localization of startegic points for surveillance and control such as units and markets that are the primary importers and ports of entry. Unfortunately the dataset available to us did not contain such information, thus limiting the scope of our study to national level movements. This should be considered for any future and expanded study of swine movements in Argentina.

Argentina has a broadly two system swine sectors with a very connected centralized industrial core conducting most movements and a very decentralized small-scale sector. Both sectors do have contacts between each other and the presence of small, but highly connected markets provides key locations to be chosen as strategic points for surveillance and control, as well as ideal places to conduct outreach to farmers about biosecurity measures and best management practices for risk-mitigation strategies.

## Author Contributions

**Conceptualization:** Jerome N. Baron, Beatriz Martínez-López.

**Data curation:** Maria N. Aznar, Mariela Monterubbianesi.

**Formal analysis:** Jerome N. Baron.

**Investigation:** Maria N. Aznar, Beatriz Martínez-López.

**Methodology:** Jerome N. Baron, Beatriz Martínez-López.

**Project administration:** Beatriz Martínez-López.

**Resources:** Beatriz Martínez-López.

**Software:** Jerome N. Baron.

**Supervision:** Maria N. Aznar, Beatriz Martínez-López.

**Validation:** Jerome N. Baron, Beatriz Martínez-López.

**Visualization:** Jerome N. Baron.

**Writing – original draft:** Jerome N. Baron.

**Writing – review & editing:** Jerome N. Baron, Maria N. Aznar, Beatriz Martínez-López.

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
