## [Decision Letter · Decision Letter 0]

14 Jan 2020

PONE-D-19-34431

Application of network analysis and cluster analysis for better prevention and control of swine diseases in Argentina

PLOS ONE

Dear Dr Baron,

Thank you for submitting your manuscript to PLOS ONE. After careful consideration, we feel that it has merit but does not fully meet PLOS ONE’s publication criteria as it currently stands. Therefore, we invite you to submit a revised version of the manuscript that addresses the points raised during the review process.

We would appreciate receiving your revised manuscript by Feb 28 2020 11:59PM. To enhance the reproducibility of your results, we recommend that if applicable you deposit your laboratory protocols in protocols.io, where a protocol can be assigned its own identifier (DOI) such that it can be cited independently in the future. For instructions see: http://journals.plos.org/plosone/s/submission-guidelines#loc-laboratory-protocols

We look forward to receiving your revised manuscript.

Kind regards,

Grzegorz Woźniakowski, PhD ScD

Academic Editor

PLOS ONE

Journal Requirements:

3. We note that Figure(s) 3 and 11 in your submission contain [map/satellite] images which may be copyrighted. All PLOS content is published under the Creative Commons Attribution License (CC BY 4.0), which means that the manuscript, images, and Supporting Information files will be freely available online, and any third party is permitted to access, download, copy, distribute, and use these materials in any way, even commercially, with proper attribution. For these reasons, we cannot publish previously copyrighted maps or satellite images created using proprietary data, such as Google software (Google Maps, Street View, and Earth). For more information, see our copyright guidelines: http://journals.plos.org/plosone/s/licenses-and-copyright.

1.    You may seek permission from the original copyright holder of Figure(s) 3 and 11 to publish the content specifically under the CC BY 4.0 license.

Additional Editor Comments (if provided):

Reviewers' comments:

Reviewer's Responses to Questions

**Comments to the Author**

1. Is the manuscript technically sound, and do the data support the conclusions?

Reviewer #1: Yes

Reviewer #2: Yes

2. Has the statistical analysis been performed appropriately and rigorously? 

Reviewer #1: Yes

Reviewer #2: Yes

3. Have the authors made all data underlying the findings in their manuscript fully available?

Reviewer #1: Yes

Reviewer #2: Yes

4. Is the manuscript presented in an intelligible fashion and written in standard English?

Reviewer #1: Yes

Reviewer #2: Yes

5. Review Comments to the Author

Reviewer #1: The paper presents very important data regarding to Argentina pigs industry - number of herds, size of herds and all possible connection between them even sesonal data. The statistical data idicate the future solution in preventing the spread of pig diseases such as African swine fever.

Line 60-63 - the sentence is to long and unclear, you should use two separete sentences.

In line 67-68, Porcine Reproductive Respiratory Syndrome, African Swine Fever and Classical Swine Fever should be written with lowercase letters

Line 77-80 you are using twice the word "re-emerging".

Line 386 - you skip comma in the number.

Line 469 - "an" not "a".

When you one use full name of the disease (African swine fever line 68) in the rest part of manuscript use shortcut (see line 471).

In reference you should use one form of ending in order to numer of pages: „p.1-2”, pp.1-2” „p. 1-2”.

In reagarding to point 3 "Have the authors made all data underlying the findings in their manuscript fully available?" - the data presented in paper are clear and available in the manuscript however the authors made declaration that the data are partly confidence:

"Data cannot be shared publicly due to confidentiality issue as this is government individual census data"

so I am not sure if the manusrcipt could be publish.

Reviewer #2: Major issues:

This is a well-written study providing insights for of swine movements networks in Argentina in order to define the most strategic points for infectious diseases prevention and control. This detailed analysis was based on comprehensive and reliable dataset, obtained from national registry of pig movements. The methods selected by the authors (social network analysis – SNA and graph theory) were previously applied by others to characterize swine movement networks in other countries, but this report was the first work regarding this issue in Argentina, thus it seems to be important and necessary in regards to the control of swine pathogens spreading.

I believe this manuscript has a great potential for publication. The manuscript is well organized and the methods are sound. The study cites current literature, which is properly placed in the context. The methods used in the study are clearly stated, the details of the methodology are sufficient to reproduce the study by other authors. The study generated a lot of data, which is presented in the tables and figures, properly placed in the manuscript. The interpretation of results is fully supported by the data, followed by comprehensive discussion with regards to similar studies performed by other authors. Moreover, also limitations of the analysis are well discussed, highlighting the need for collecting of more data regarding individual pig movement to improve the resolution of the study (lines558-559). The use of sample farms A, B, C in discussion significantly improved understanding of the limitations of the study. Discussion section is well written, nevertheless the context of infectious diseases spreading in the context of obtained data is slightly insufficient and should be expanded to reinforce the meaning of the obtained results. The study is performed on the country level, but in the context of exotic diseases, also the issue of pig import into Argentina should be at least mentioned in the introduction, and need minor discussion on the background of obtained data. Any information of pig of foreign origin on the market might indicate the potential sources of disease introduction into Argentina. Nevertheless, this minor missing issues did not affect overall high quality and informativeness of the study.

Minor issues:

1. If is is possible, the tables 2-7 should be moved into supplementary information.

2. White background in the figures 1 and 2 will improve graphical presentation of the data.

3. Figure 10 is not necessary, depicted node is easy to observe at figure 11.

4. Lines 73-75: reference is missing.

5. Lines 77-80: “[…]transboundary, re-emerging diseases if they enter the country” – remove unnecessary comma and change “should” to “if”.

6. Lines 87-88: comma should stand instead of full stop after reference no 13.

7. Line 469: an important

8. 67-68, 471-472: diseases names should be written lowercase, except “African swine fever”.

9. Line 479-480: I would change “[…]we have what is known as the[…]” into “there is so-called”

10. Lines 483-486: too many words of “small”, try to use synonymous words

11. Line 584: “[…]would be the most useful”

6. PLOS authors have the option to publish the peer review history of their article (what does this mean?). If published, this will include your full peer review and any attached files.

Reviewer #1: No

Reviewer #2: No

---

## [Author Response · Author response to Decision Letter 0]

7 May 2020

Dear editors

We thank you for your comments. Please find attached our response to the reviewers’ comments to our manuscript entitled “Application of network analysis and cluster analysis for better prevention and control of swine diseases in Argentina” by Dr. Jerome Baron, Dr. Maria Aznar, Dr. Mariela Monterubbianesi and Dr. Beatriz Martinez Lopez, as well as an edited copy taking into account these comments. We hope we have addressed the comments as to make this manuscript suitable for publication.

Thank you in advance for your consideration,

Best Regards,

Jerome Baron, DVM, MSc

Center for Animal Disease Modeling and Surveillance (CADMS)

Department of Medicine & Epidemiology

School of Veterinary Medicine

University of California

Davis, CA 95616 USA

jnbaron@ucdavis.edu

APRIL 28 2020 REVIEW

1. Please amend the manuscript submission data (via Edit Submission) to include author Mariela Monterubbianesi.

Author has been added

2. Thank you for taking careful note of Google Map’s policies--their license on map images indeed does not comply with the license PLOS uses, CC BY 4.0 (https://creativecommons.org/licenses/by/4.0/). To confirm that the sources of your new map images do comply with our policy, we still require some additional information. Please indicate what “source and package” you used to create the images and the “open sources shape files.”

The R package used for map sourcing (package “map”) was already referenced (reference 16). The data source (Natural Earth data) for the Argentina shapefile has been added (reference 17), and is indeed open-sourced as is specified in their terms of use (https://www.naturalearthdata.com/about/terms-of-use/)

3. We note our Data Availability Statement reads: “No - some restrictions will apply. Data cannot be shared publicly as this data is owned by a third-party and has confidentiality issue as this is individual census data. Data accessibility and restriction information can be obtained from the National Institute of Agricultural Technology (INTA) and the National Service of Agri-Food health and Quality (SENASA). For more information about data accessibility please contact infopublica@senasa.gob.ar.”

Before we proceed with the review process, we’ll require some additional information to ensure your submission adheres to the PLOS ONE policy regarding acceptable third-party data restrictions: https://journals.plos.org/plosone/s/data-availability#loc-acceptable-data-access-restrictions.

1.) Please confirm that the authors had no special access privileges to the data and that other researchers will be able to access the data in the same manner as the authors.

I do confirm

2.) Please confirm whether access requests for both the INTA and SENASA data can be sent to infopublica@senasa.gob.ar. If not, please provide non-author contact information (preferably email) to which INTA data access requests can be sent.

I do confirm that data requests can be made at this email. However, I made a mistake in the original statement, data belongs to SENASA only and not INTA. I have modified the statement in the online submission page to correct this and address the 2 requests.

JANUARY 13 2020 REVIEW

We have made sure that all template requirements have been to the extant of our observations.

Data are owned by a third-party organization, the Argentine National Service of Agri-Food Health (SENASA), a department of the Ministry of Agriculture, which doesn’t not allow us to share the data directly. These data are collected for SENASA’s surveillance operations.They contain sensitive and identifiable information regarding this country’s production system and individual farmers. Moreover, given the analysis completed in this paper, fully de-indentifying the data would involve removing spatial coordinates, which would not make our findings reproducible, as they involved spatial methods that used the detailed individual locations of farms. Researchers may ask about data availability and restrictions to SENASA directly, here is the contact info:

infopublica@senasa.gob.ar

3. We note that Figure(s) 3 and 11 in your submission contain [map/satellite] images which may be copyrighted. All PLOS content is published under the Creative Commons Attribution License (CC BY 4.0), which means that the manuscript, images, and Supporting Information files will be freely available online, and any third party is permitted to access, download, copy, distribute, and use these materials in any way, even commercially, with proper attribution. For these reasons, we cannot publish previously copyrighted maps or satellite images created using proprietary data, such as Google software (Google Maps, Street View, and Earth). For more information, see our copyright guidelines: http://journals.plos.org/plosone/s/licenses-and-copyright.

In searching for the possibility of obtaining copyright access from Google maps to use their background, we found a statement from them stipulating that they not grant explicit written permission for use of their content, though use of their maps is still permitted. This is stated in the link below:

https://www.google.com/permissions/geoguidelines/

Thus to remove doubt, we remade the maps using a new source and package with no copyright issues as the figure now uses open-sourced shapefiles. References and legends have been updated accordingly

Reviewer #1: The paper presents very important data regarding to Argentina pigs industry - number of herds, size of herds and all possible connection between them even sesonal data. The statistical data idicate the future solution in preventing the spread of pig diseases such as African swine fever.

Line 60-63 - the sentence is to long and unclear, you should use two separete sentences.

Changed as suggested (line 60-65)

In line 67-68, Porcine Reproductive Respiratory Syndrome, African Swine Fever and Classical Swine Fever should be written with lowercase letters

Changed as suggested (line 69-70)

Line 77-80 you are using twice the word "re-emerging".

Changed as suggested (line 81)

Line 386 - you skip comma in the number.

Changed as suggested (line 388)

Line 469 - "an" not "a".

Changed as suggested (line 472)

When you one use full name of the disease (African swine fever line 68) in the rest part of manuscript use shortcut (see line 471).

Changed as suggested (line 474, 475)

In reference you should use one form of ending in order to numer of pages: „p.1-2”, pp.1-2” „p. 1-2”.

Changed to p.1-2 format (ref 1, 2, 5, 8, 10, 11)

In reagarding to point 3 "Have the authors made all data underlying the findings in their manuscript fully available?" - the data presented in paper are clear and available in the manuscript however the authors made declaration that the data are partly confidence:

"Data cannot be shared publicly due to confidentiality issue as this is government individual census data"

so I am not sure if the manusrcipt could be publish.

We meant that the detailed dataset with individual observations, location and identification numbers was confidential due to the need to protect individual swine operations and the fact that this data is proprietary to SENASA. The summarized data as presented in the manuscript has been approved for publication by our collaborators.

Reviewer #2: Major issues:

This is a well-written study providing insights for of swine movements networks in Argentina in order to define the most strategic points for infectious diseases prevention and control. This detailed analysis was based on comprehensive and reliable dataset, obtained from national registry of pig movements. The methods selected by the authors (social network analysis – SNA and graph theory) were previously applied by others to characterize swine movement networks in other countries, but this report was the first work regarding this issue in Argentina, thus it seems to be important and necessary in regards to the control of swine pathogens spreading.

I believe this manuscript has a great potential for publication. The manuscript is well organized and the methods are sound. The study cites current literature, which is properly placed in the context. The methods used in the study are clearly stated, the details of the methodology are sufficient to reproduce the study by other authors. The study generated a lot of data, which is presented in the tables and figures, properly placed in the manuscript. The interpretation of results is fully supported by the data, followed by comprehensive discussion with regards to similar studies performed by other authors. Moreover, also limitations of the analysis are well discussed, highlighting the need for collecting of more data regarding individual pig movement to improve the resolution of the study (lines558-559). The use of sample farms A, B, C in discussion significantly improved understanding of the limitations of the study. Discussion section is well written, nevertheless the context of infectious diseases spreading in the context of obtained data is slightly insufficient and should be expanded to reinforce the meaning of the obtained results. The study is performed on the country level, but in the context of exotic diseases, also the issue of pig import into Argentina should be at least mentioned in the introduction, and need minor discussion on the background of obtained data. Any information of pig of foreign origin on the market might indicate the potential sources of disease introduction into Argentina. Nevertheless, this minor missing issues did not affect overall high quality and informativeness of the study.

As import/export data was not made available to us, we had to limit the scope of our study at the national level. We are aware that imports of foreign pigs are a potential source for introduction of new diseases. This was addressed in a new paragraph in the conclusion (line 652-658).

Minor issues:

1. If is is possible, the tables 2-7 should be moved into supplementary information.

We think tables 2-7 are key for the understanding and reference of the paper results and, therefore should be kept in the main text, not as supplementary information.

2. White background in the figures 1 and 2 will improve graphical presentation of the data.

Changed as suggested

3. Figure 10 is not necessary, depicted node is easy to observe at figure 11.

Figure removed and figure references adjusted accordingly (lines 378-380, 394-395, 401, 402, 405, 408, 432, 438, 441, 450, 460, 637)

4. Lines 73-75: reference is missing.

Added a reference as suggested (lines 76-77, ref 4)

5. Lines 77-80: “[…]transboundary, re-emerging diseases if they enter the country” – remove 

unnecessary comma and change “should” to “if”.

Changed based on reviewer 1’s comment

6. Lines 87-88: comma should stand instead of full stop after reference no 13.

Changed as suggested (line 90)

7. Line 469: an important

Changed as suggested (line 472)

8. 67-68, 471-472: diseases names should be written lowercase, except “African swine fever”.

Changed as suggested (line 69-70)

9. Line 479-480: I would change “[…]we have what is known as the[…]” into “there is so-called”

Sentence modified to fit suggestion (line 482-484)

10. Lines 483-486: too many words of “small”, try to use synonymous words

Sentence changed with new worthing (line 488-489)

11. Line 584: “[…]would be the most useful”

Changed as suggested (line 588)

---

## [Decision Letter · Decision Letter 1]

28 May 2020

Application of network analysis and cluster analysis for better prevention and control of swine diseases in Argentina

PONE-D-19-34431R1

Dear Dr. Baron,

We are pleased to inform you that your manuscript has been judged scientifically suitable for publication and will be formally accepted for publication once it complies with all outstanding technical requirements.

With kind regards,

Grzegorz Woźniakowski, PhD ScD

Academic Editor

PLOS ONE

Additional Editor Comments (optional):

Reviewers' comments:

Reviewer's Responses to Questions

**Comments to the Author**

1. If the authors have adequately addressed your comments raised in a previous round of review and you feel that this manuscript is now acceptable for publication, you may indicate that here to bypass the “Comments to the Author” section, enter your conflict of interest statement in the “Confidential to Editor” section, and submit your "Accept" recommendation.

Reviewer #1: All comments have been addressed

Reviewer #2: All comments have been addressed

2. Is the manuscript technically sound, and do the data support the conclusions?

Reviewer #1: (No Response)

Reviewer #2: Yes

3. Has the statistical analysis been performed appropriately and rigorously? 

Reviewer #1: (No Response)

Reviewer #2: Yes

4. Have the authors made all data underlying the findings in their manuscript fully available?

Reviewer #1: (No Response)

Reviewer #2: Yes

5. Is the manuscript presented in an intelligible fashion and written in standard English?

Reviewer #1: (No Response)

Reviewer #2: Yes

6. Review Comments to the Author

Reviewer #1: (No Response)

Reviewer #2: (No Response)

7. PLOS authors have the option to publish the peer review history of their article (what does this mean?). If published, this will include your full peer review and any attached files.

Reviewer #1: No

Reviewer #2: No

---

## [Editor Report · Acceptance letter]

3 Jun 2020

PONE-D-19-34431R1 

Application of network analysis and cluster analysis for better prevention and control of swine diseases in Argentina 

Dear Dr. Baron:

I'm pleased to inform you that your manuscript has been deemed suitable for publication in PLOS ONE. Congratulations! Your manuscript is now with our production department. 

Kind regards, 

on behalf of

Prof. Grzegorz Woźniakowski 

Academic Editor

PLOS ONE